

# Distribution and degradation of terrestrial organic matter in the
# sediments of peat-draining rivers, Sarawak, Malaysian Borneo
Ying Wu[1], Kun Zhu[1], Jing Zhang[1], Moritz Müller[2], Shan Jiang[1], Aazani Mujahid[3], Mohd Fakharuddin
Muhamad[3], Edwin Sien Aun Sia [2]
1 State Key Laboratory of Estuary and Coastal Research, East China Normal University, Shanghai,
China.
2 Faculty of Engineering, Computing and Science Swinburne, University of Technology, Sarawak
campus, Malaysia.
3 Faculty of Resource Science & Technology, University Malaysia Sarawak, Sarawak, Malaysia.
Correspondence,
Ying Wu, wuying@sklec.ecnu.edu.cn



## Abstract.

Tropical peatlands are one of the largest pools of terrestrial organic carbon (OCterr); however, our understanding of the dynamics of OCterr in peat-draining rivers remains limited, especially in Southeast Asia. This study used bulk parameters and lignin phenols concentrations to investigate the characteristics of OCterr in a tropical peat-draining river system (the main channel of the Rajang and three smaller rivers) in the western part of Sarawak, Malaysian Borneo. The depleted $\delta^{13}C$ levels and lignin composition of the organic matter indicates that the most important plant source of the organic matter in these rivers is woody angiosperm C3 plants, especially in the three small rivers sampled. The diagenetic indicator ratio (i.e., the ratio of acid to aldehyde of vanillyl phenols (Ad/Al)v) increased with decreasing mean grain size of sediment from the small rivers. The selective sorption of acid relative to aldehyde might explain the variations in the (Ad/Al)v ratio. The (Ad/Al)v ratio appears to be related to the C/N ratio (the ratio of total organic carbon to total nitrogen) in the Rajang and small rivers, where slower degradation of OCterr and a higher total nitrogen percentage (TN%) were observed, compared to other river systems. Most of the OCterr discharged from the Rajang and small river systems was material derived from woody angiosperm plants with limited diagenetic alteration before deposition, and so could potentially provide significant carbon to the atmosphere after degradation.

## 1 Introduction

Tropical peatlands are one of the biggest terrestrial organic carbon pools, accounting for about 89,000 Tg (Moore et al., 2013; Rieley et al., 1996, 2008). It is reported that about 77% of the carbon stored in all tropical peatlands derived from Southeast Asia, which equals to 11%–14% of the total carbon pool stored in all peat. However, increasing anthropogenic disturb in the form of land use change, drainage and biomass burning are converting this peat into a globally significant source of atmospheric carbon dioxide (Dommain et al., 2014; Miettinen et al., 2016;



42 Koh et al., 2009; Page et al., 2011). The rivers draining these peatlands are typically rich in lignin

43 phenols and humic substances, and are often referred to as "blackwater" rivers (Baum et al., 2007;

44 Cook et al., 2017; Moore et al., 2011). However, knowledge of the fate of terrigenous organic

45 matter in such peat-draining rivers and estuaries remains limited (Gandois et al., 2014; Hall et al.,

46 2015; Lourençato et al., 2019).

47 The transport, degradation, and sequestration of OCterr in river systems are important because

48 of their roles in constraining carbon cycle budgets (Aufdenkampe et al., 2011; Battin et al., 2009;

49 Feng et al., 2016; Spencer et al., 2010; Wu et al., 2018). In terms of transport within fluvial systems,

50 OCterr is subject to various natural processes, such as photo bleaching, microbial degradation,

51 and selective preservation, as well as anthropogenic activities e.g. dam construction, irrigation

52 systems, and land use change (Bao et al., 2015; Hernes et al., 2017; Spencer et al., 2010; Wu et al.,

53 2015, 2018). Thus, it can be difficult to distinguish OCterr behavior from dynamics within a fluvial

54 system. Multiple geochemical approaches have been applied to elucidate the composition and

55 fate of OCterr in riverine and coastal sediments, including C/N ratios, $\delta^{13}C$ composition, and the

56 distribution and composition of specific biomarker compounds such as lignin phenols and plant

57 wax n-alkanes (Bao et al., 2015; Drenzek et al., 2007; Goni et al., 2005; Hernes and Benner, 2002;

58 Jex et al., 2014; Ward et al., 2013). Lignin, which constitutes up to 30% of vascular plant biomass,

59 is a unique biomarker of OCterr (Goñi and Hedges, 1995; Hedges and Mann, 1979). The

60 monomeric composition of lignin phenols (S, V, C series) provides useful information on the

61 biological source (woody *versus* nonwoody and angiosperm *versus* gymnosperm) and oxidation

62 stage of lignin in natural environments (Benner et al., 1984; Hedges et al., 1985; Dittmar and Lara,

63 2001; Tareq et al., 2004; Thevenot et al., 2010). Most studies designed to understand the sources,

64 compositions and transport of exported OCterr to determine its impact on the carbon cycle have

65 been carried out in large rivers in the temperate and polar zones (Bao et al., 2015; Bianchi et al.,

66 2002, 2011; Drenzek et al., 2007; Goñi et al., 1998, 2005; Feng et al., 2016; Wu et al., 2015, 2018).

67 In contrast, lignin signatures from tropical environments have received less attention, especially

68 in small river systems (Alin et al., 2008; Alkhatib et al., 2007; Dittmar and Lara, 2001; Goñi et al.,



2006; Hedges et al. 1986; Spencer et al., 2010; Sun et al., 2017; Pradhan et al., 2014).
The export of OCterr in tropical river systems is typically constrained by natural rainfall, typhoons,
floods, and tectonic activity (Alin et al., 2008; Aufdenkampe et al., 2007; Bao et al., 2015). Elevated
soil turnover rates, coupled with short water residence times in small tropical river catchments,
lead to the accelerated transformation of terrestrial organic matter (OM), especially during high-
discharge events (Bao et al., 2015; Goldsmith et al., 2008; Kao and Liu, 1996). Anthropogenic
processes such as deforestation have been a major cause of altered hydrology and OM
compositions in tropical river systems (Houghton et al., 2000; Jennerjahn et al., 2004, 2008;
Pradhan et al., 2014). The current paucity of information on OCterr characteristics and its export
by rivers from tropical peat-draining rivers remains a major gap in our understanding of OCterr
biogeochemical cycling in rivers from tropical Southeast Asia. Previous studies have reported
that peatland-draining rivers in Sumatra and Borneo contained the highest values of dissolved
organic carbon (DOC) in rivers globally (3000–5500 µmol L$^{-1}$), and most of the terrestrial DOC
delivered into the sea (Wit et al., 2015). To understand the biogeochemical processing of OCterr
in Southeast Asia, more work is needed on the dynamics of OCterr in the fluvial systems of this
region.
Here we present what is, to our knowledge, the first analysis of OCterr concentration and
behavior in four rivers and estuarine regions in the western part of Sarawak, Malaysian Borneo.
We examined the OCterr characteristics using the lignin phenols composition from various
samples (e.g., plants, soils, and sediments) from a major river, the Rajang, and three adjacent
small rivers (the Maludam, Simunjan, and Sebuyau) to resolve the sources and transformation
processes in the wet *versus* dry season. We further compared data among the four rivers to
determine the ultimate fate of lignin and the potential controls on its distribution. Our results
also indicate that lignin composition links to sources and modifications along the river–peat/soil–
estuary continuum and reveal its response to peat degradation.

## 2 Materials and methods





## 2.1 Study region and sample collection


Samples were collected during three field expeditions to Sarawak in August 2016 (only the
Rajang), early March 2017 (the Rajang and the three small rivers), and September 2017 (only the
small rivers; Fig. 1). During the 2017 expeditions, typical plants (Table S2) and soil samples were
also collected for the comparison study.
The Rajang River drainage basin covers an area of about 50,000 km$^2$. Elevations exceed 2000 m
and hill slopes are steep, generally in excess of 258 m in the interior highlands and 208 m in
lower areas (Martin et al., 2018). The three small rivers (the Maludam, Simunjan, and Sebuyau)
are blackwater rivers that draining extensive peatlands (Fig. 1). For the Rajang, it is separated into
two parts by Sibu Town, upper reaches mainly drains mineral soils, while down reaches develops
multiple distributary channels (e.g., the lower Rajang, Serendeng, Igan; Fig. 1). These channels
are also surrounded by broad peatlands (Staub et al., 2000). However, Deforestation and
changing in land use are accelerating the peatland degradation (Fig. 1). More than 50%
peatland (11% of the catchment size) in Rajang watershed has been occupied by industry
plantation (e.g. oil palm) (Miettinen et al., 2016). Fishery, logging and timber processing are the
traditional supports for local citizens (Miettinen et al., 2016).
The climate of the study area is classified as tropical ever-wet, with average rainfall in excess of
3700 mm/year. The average monthly water discharge is about 3600 m$^3$/s, with peak discharge
during the northeastern monsoon season (December to March; Staub et al., 2000). The three
sampling periods resembled the end of this northeastern monsoon (i.e., March, the end of the
wettest season of the year) and were shortly before the beginning of the northeastern monsoon
(i.e., August and September, the end of the drier season).
The surface sediments were collected using grab samplers from a small boat at each station and
then 0 - 5 cm subsamples were removed and frozen (–20°C) until they were dried for subsequent
analysis in the laboratory. The dominant botanical samples and soils within the basin were
collected at the same time and stored in a freezer. The hydrological parameters of the surface
river water (e.g., salinity, pH, and temperature) at each station were determined using an





Aquaread® multiple parameters probe (AP-2000).

**2.2 Chemical analysis**

Prior to chemical analysis, all botanical samples as well as the soil and sediment samples, were
dried at 55 °C and disaggregated in an agate mortar to form a homogeneous sample.
Grain size characteristics were measured directly from aliquots of the surface sediment samples
using a Coulter LS 100Q (Coulter Company, USA), after treatment with 5% $H_2O_2$ and 0.2M HCl
to dissolve organic matter and biogenic carbonate. The sediment grain sizes are expressed as
the proportions of clay (<4 μm), silt (4–63 μm), and sand (>63 μm), with a measurement error
of ≤5% for the entire dataset. The remaining sediments were ground to 80 mesh (187.5 μm) for
elemental, isotopic, and lignin analyses.
The concentrations of organic carbon and total nitrogen (TN) were analyzed using a CHNOS
Elemental Analyzer (Vario EL III) with a relative precision of ±5%. The weight percentages of
organic carbon were analyzed after removing the carbonate fraction by vapor phase acidification.
The weight percentages of TN were also analyzed following the same procedure but without
acidification. The stable carbon isotopic composition of the decarbonated sediments was
determined by a Flash EA1112 Elemental Analyzer connected to an Isotope Ratio Mass
Spectrometer (MAT Delta Plus/XP, Finnigan). $^{13}C/^{12}C$ ratios are expressed relative to the PDB
standard using conventional δ notation. The analytical precision, determined by replicate analysis
of the same sample, was ±0.2‰.
Lignin phenols were extracted using the cupric oxide digestion technique (CuO; Hedges and
Ertel, 1982; Yu et al., 2011). Briefly, the powdered samples were weighed and placed in $O_2$ free
Teflon-lined vessels, and digested in a microwave (CEM MARS5) at 150°C for 90 min (Goñi and
Montgomery, 2000). Samples were then acidified to pH < 2 and phenolic monomers were
extracted into 99:1 (volume ratio) ethyl acetate/petroleum ether, dried, and stored at −20°C until
further analysis. Samples were analyzed as trimethylsilyl derivatives of N,O-
bis(trimethylsilyl)trifluoroacetamide (BSTFA) and trimethylchlorosilane (TMCS; 99:1) by Agilent



6890N gas chromatography (DB–1 column, FID). The lignin phenols concentration were
quantified using calibration curves based on commercial standards (Sigma Aldrich). Eleven
phenol monomers were extracted and categorized into five groups: syringyl (S), vanillyl (V),
cinnamyl (C), p-hydroxyl (P), and 3,5-dihydroxy benzoic acid (DHBA).
These phenol monomers are detected by gas chromatography in their trimethylsilated forms.
The S, V, and P groups further contain three monomers with aldehyde (–CHO), ketone (C=O),
and carboxylic acid ($-CO_2H$) functional groups. Any complications are usually related to
conventional diagenesis indicators such as the S/V and cinnamyl/vanillyl phenols (C/V) ratios
(see Table 1), and these were also solved by normalizing V, S, and C to total lignin phenols as
defined below:
Lignin phenols vegetation index (LPVI) = [{S(S + 1)/(V + 1) + 1} × {C(C + 1)/(V + 1) + 1}]

**2.3 Statistical analysis**
All statistical analyses were carried out using SPSS 10.0 (IBM SPSS Inc., USA) and results were
plotted using Origin software (Origin Lab Inc., USA). Multivariate statistical approaches such as
principle component analysis (PCA) and cluster analysis (CA) are among the most widely used
statistical methods in determining the significance of specific parameters within a dataset
(Pradhan et al., 2009). Interrelationships among the sampling points in different rivers were
characterized by cluster analysis using Ward's method (linkage between groups) and similarity
measurements in terms of Euclidian distance, illustrated in dendograms.

**3 Results**
**3.1 Hydrological parameters, grain size, and bulk elemental and stable isotopic composition of**
**vegetation, soil, and sediment**
The hydrological parameters for the study area are summarized in Table S1. The salinity of the
lower Rajang system varied significantly (from 12‰ to 32‰) because of saline water intrusion
in the estuarine region, but there were limited pH variations (6.5–7.9). Dissolved oxygen (DO)





levels show significant spatial variations, with the lowest values (2–3 mg L⁻¹) being recorded in
the Igan channel, where dense peats were observed, and the higher values (4–6 mg L⁻¹) recorded
in the other two channels. The salinity of the Simunjan indicates that freshwater dominated,
whereas the two other small rivers showed saline water influences. The variation in pH values
among the three small rivers decreased from the Sebuyau (~6.4), to the Simunjan (~5.1), and the
Maludam (~3.7). The DO concentrations in the three small rivers varied in a low range (average:
2–3 mg L⁻¹), with the lowest values in the three systems being around 1.4 mg L⁻¹.
The compositions of bulk sediments from the Rajang and the three small rivers are presented in
Tables 1 and S1. The mean grain sizes from the upper Rajang (212±47 μm) are much coarser than
those from the lower Rajang (40±38 μm) and the small rivers (22±16 μm). The finest samples
(9±2 μm) were collected from the Maludam in March 2017. Generally, the samples collected
during the dry season were coarser than those from the flood season in the Maludam and
Simunjan, but this was not the case for the Sebuyau. The average organic carbon content shows
a significant negative relationship with mean grain size among these samples ($r^2$ = 0.67, p < 0.01).
Mean values of Total organic carbon (TOC) concentrations were higher in the peat-draining
rivers (2.2±0.58%, 2.6±1.23%, and 2.6±0.8% for the Maludam, Sebuyau, and Simunjan,
respectively) compared with the lower Rajang (1.1±0.5%), and the lowest values were observed
in the upper Rajang (0.12±0.02%). The highest values of OC for the vegetation were measured
in plants samples (30%–49%). The mean TOC value in the soil samples was 3.6±0.6%.
TN content ranged from 0.02% to 0.17% in the samples collected from the Rajang, from 0.09%
to 0.37% in the small rivers, from 0.73% to 1.65% in the vegetation, and averaged 0.19±0.02%
for the soil samples (Tables 1, S2, and S3). Although nitrogen was enriched in the samples from
the peat-draining rivers, they still had higher mean C/N values (15.8±3.7) compared with the
lower Rajang (11.5±1.6) while vegetation samples, which exhibited low N content and high C/N
(C/N = 56±34).
The most abundant vegetation collected from the Maludam showed relatively depleted carbon
isotope ratios (δ¹³C = −31‰) that are typical of C3 vegetation (Table S2). The detritus samples





were also relatively depleted in $^{13}$C ($\delta^{13}$C = −29.2‰; Table 1). The isotope ratio of the peat-
draining river samples was slightly enriched in $^{13}$C (average $\delta^{13}$C = −28.0±0.4‰) compared with
the Rajang (average $\delta^{13}$C = −28.7±0.6‰). The $\delta^{13}$C values of the soil samples are similar to those
of the small rivers ($\delta^{13}$C = −28.4‰).

### 3.2 Lignin phenols content
The lignin phenols obtained after CuO oxidation are expressed as $\Lambda 8$ (mg (100 mg OC)$^{-1}$) ,
except for the lignin yield ($\Sigma 8$), which is the sum of C + S + V and is expressed as mg 10 mg dw$^{-1}$
, and are presented in Fig. 2 as well as Tables 2 and S1-3. The highest yields were measured in
the vegetation samples (300–900 mg 10 mg dw$^{-1}$). The lignin yield from the soil samples and the
three small rivers (average of ~30 mg 10 mg dw$^{-1}$) is also higher than that from the Rajang
samples (average of <10 mg 10 mg dw$^{-1}$), with the lowest value observed in the upper Rajang
(0.16 mg 10 mg dw$^{-1}$; Table 2). There are good correlations between $\Sigma 8$ and OC% in each river,
with the slope decreasing in the order of Maludam > Simunjan > Sebuyau > Rajang (Fig. 2a).
The variation in $\Lambda 8$ from various pools shows a similar distribution as the $\Sigma 8$ values. The average
concentrations for the vegetation, soil, and the four river systems mg (100 mg OC)$^{-1}$
approximately 18, 8.3, 5.4 mg (100 mg OC)$^{-1}$ (for the Rajang), 6.2 mg (100 mg OC)$^{-1}$ (for the
Maludam), 7.9 (for the Sebuyau), and 7.4 mg (100 mg OC)$^{-1}$ (for the Simunjan), respectively.
The C/V and S/V ratios differ with vegetation type (Fig. 2b). Angiosperm leaves show high S/V
(>1) and C/V ratios (~0.8). Angiosperm wood and root samples show lower C/V ratios (<0.2).
The detritus samples show intermediate S/V ratios (0.6–1.0) and lower C/V ratios (~0.1). Soil
samples have relatively high S/V (~1.1) and low C/V (~0.07) values. The four rivers show limited
variations in S/V (0.4–0.8) and C/V (0.02–0.08) ratios. The LPVI values of the fresh plant material
range from 113 to 2854 for leaves and 192 to 290 for wood. The values for detritus range between
36 and 228, and for soil and sediment range between 30 and 60 (Table 2).
The ratios of vanillic acid to vanillin ($(Ad/Al)_V$) and syringic acid to syringaldehyde ($(Ad/Al)_S$)
increase slightly from the vegetation to river samples (Table 2). The ratios obtained for the





vegetation and soil samples show similar values ($(Ad/Al)_S$ = ~0.30; $(Ad/Al)_V$ = ~0.35). The ratios
from the small river samples range from 0.41 to 0.58 for $(Ad/Al)_V$ and 0.30 to 0.36 for $(Ad/Al)_S$.
The values from the lower Rajang are similar to those from the small rivers, but this is not the
case for the upper Rajang, where higher $(Ad/Al)_S$ and $(Ad/Al)_V$ values were recorded. The two
ratios are linearly correlated in all sediment samples ($r^2$ = 0.68, p < 0.05), except for the samples
collected from the Simunjan.
The P/(V + S) ratio is low in the vegetation samples, except for the leaf samples (P/(V + S) = 0.22),
which reflects the very low V content (Table 2). The ratio is 0.28±0.03 for the soil samples,
0.18±0.4 for the small rivers, 0.17±0.02 for the lower Rajang, and 0.51±0.04 for the upper Rajang.
The 3,5-dihydroxybenzoic (DBHA) levels determined from the soil and sediments are plotted in
Fig. 2d. DHBA is very low in the upper Rajang (~0.07), but higher in the Maludam in the dry
season (average value of 0.44). Values in the Simunjan in both seasons are similar to those from
the soil samples (~0.38). Higher values of DHBA were measured in the lower Rajang and the
Sebuyau in the dry season than in the wet season.

**3.3 Statistical analysis**
The results of cluster and PCA analyses of both bulk geochemical and lignin phenols proxies for
all sediments are shown in Fig. 3. Four distinct groups were identified based on the cluster
analysis. The Maludam and the tributary of the lower Rajang (Igan) are grouped together, and
the Simunjan and Sebuyau are grouped together. The lower Rajang and upper Rajang are
separated from each other (Fig. 3a). Similar groupings are evident in the results of the PCA
analysis, which was based on the distribution of factors 1 and 2 that represent total loadings of
45% and 32%, respectively (Fig. 3b). The PCA results show a close relationship between Σ8 and
OC% in factor 2, whereas the $(Ad/Al)_V$ ratio is related to grain size in factor 1.

**4 Discussion**
**4.1 Comparison with systems worldwide: lignin parameters derived from sediment and peat**





Table 3 summarizes the distribution of bulk and lignin parameters from typical systems
worldwide. Although the TOC values of our studied systems are compared lower with peat
samples but the concentrations of lignin phenols are comparable, which are typically enriched
in lignin phenols compared with other river systems (Table 3; Bianchi et al., 2002; Gandois et al.,
2014; Li et al., 2015; Sun et al., 2017; Pradhan et al., 2014; Winterfeld et al., 2015). The TN values
of our peat samples are between two and four times higher than those seen in other systems
worldwide, as was also observed in small rivers along India's west coast (Pradhan et al., 2014).
The higher values of $\Lambda 8$ found in our studied systems were potentially caused by peat-draining
and intense human activity near the watersheds, as reported previously (Milliman and Farnsworth,
2011; Moore et al., 2013; Rieley et al., 2008). Much of the peatland neighboring the Simunjan and
Sebuyau catchments has been changed to palm oil plantations (Martin et al., 2018). The
terrigenous OM has been affected by diagenesis, as $(Ad/Al)_V$ varies markedly among the
different systems (Table 3). The fluvial matter across watersheds within the Arctic region is
strongly degraded (Winterfeld et al., 2015). The $(Ad/Al)_V$ values of the sediments sampled here
are comparable to fresh and only low to medium oxidized. This study provides new insights into
the amount of terrestrial OC preserved in the tropical delta region of southeastern Borneo, as
well as into the biogeochemical transformation of OM from terrestrial source to marine sink
across this region.

**4.2 Origin of sediment organic matter in tropical peat-draining rivers**
The depleted average $\delta^{13}C$ values (–28.5‰) of our vegetation samples indicate an insignificant
contribution from C4 plants in the study area (Gandois et al., 2014; Sun et al., 2017). The high C/N
ratio (64.8) indicates a predominance of terrestrial high plant species (e.g., *Nepenthes sp.* and
*Avicennia marina Vierh.*). The $\delta^{13}C$ and C/N values (–27.2‰ and 12, respectively) obtained from
the soil and sediments collected near the rivers suggest that terrestrial organic matter is the
dominant contributor (Table 1). The enriched $\delta^{13}C$ values obtained from the peat-draining rivers
when compared with the Rajang could be the result of higher contribution of peatland



vegetation (Benner et al., 1987; Gandois et al., 2014). The cluster and PCA analyses suggest that
there were no significant seasonal differences in these rivers. This might be because of the similar
precipitation levels during our sampling seasons and sediments samples related to long-term
records compared with particulate phase (Martin et al., 2018). The close relationship between the
OC% and $\Sigma 8$ in the PCA suggests factor 2 relates to the source of the organic matter (Fig. 3), as
also indicated by the strong correlation between OC% and $\Sigma 8$ (Fig. 2). Correlation of OC% and
$\Sigma 8$ of the Maludam showed the highest slope, possibly related to its pristine condition that
promotes better conservation of vegetation in its peat. Furthermore, the differences between
the upper and lower Rajang are highlighted by the PCA results and bulk parameters; i.e., the
upper Rajang drains a mineral soil whereas peat is dominant in the delta region. This also explains
why the Rajang data do not plot with the other small river systems; the linear relationship
between $\delta^{13}C$ and $\Sigma 8$ for the Rajang ($r^2 = 0.92$) forms a distinct group separate from the small
rivers ($r^2 = 0.59$; Fig. 3).
The S/V and C/V ratios are often used as indicators of the vegetation origin of the lignin fraction;
e.g., the woody and non-woody parts of gymnosperm and angiosperms (Hedges and Mann,
1979). The S/V values (<0.8) of the peat-draining rivers are slightly lower than the values of other
peats (<1.5), but the C/V ratios are comparable (Tareq et al., 2004). The differences in these
parameters between the sediments and the vegetation and soils, as illustrated in Fig. 2, suggests
that they are composed mostly of angiosperm wood. This finding is further confirmed by the
LPVI values, which are commonly less than 60 in these sediment samples. Previous studies have
concluded that tropical peats are derived mainly from wood (Anderson, 1983; Gandois et al.,
2014). For the Rajang, the LPVI values show a positive linear correlation with $\Lambda 8$ concentrations
($r^2 = 0.56$); however, for the small rivers (based on mean values, except the samples collected in
March 2017 from the Maludam) this relationship shows a negative correlation ($r^2 = 0.91$). This
suggests that the small rivers receive more lignin derived from woody material, whereas the
Rajang has a mixture of sources. The unusual behavior of the Maludam's samples might be
related to the dominance of finer-grained sediments when compared with the other rivers,





because woody material tends to be concentrated in the coarser fraction (Table 1).
P phenols in the Rajang are derived from lignin, as supported by the significant correlation of
the content of P phenols and lignin content ($r^2$ = 0.93). However, there is no correlation between
P phenols and lignin content for the small rivers. All P/V values from the samples (0.13–0.72) are
higher than the average P/V ratio of wood (0.05) and are similar to the range observed for leaves
(0.16–6.9; Hedges et al., 1986). Considering this, non-woody angiosperms are the most likely
source of additional lignin.

**4.3 Transformation of lignin signatures in tropical peat-draining rivers**
$(Ad/Al)v$ ratios are often used to estimate the degradation status of terrestrial OM. The $(Ad/Al)v$
ratios for soils reported in previous studies fall within the range 0.16–4.36, 0.1–0.2 for fresh
angiosperm wood and 0.2–0.6 for non-woody tissues (Hedges et al., 1988; Opsahl and Benner,
1995; Thevenot et al., 2010). In our study, the variability of the $(Ad/Al)v$ ratios obtained from the
vegetation, soil, and sediments was limited, with values between 0.3 and 0.58 except from the
samples from the upper Rajang (~1.0), which suggests the mild degradation of OCterr in most
samples. The degradation status of lignin was negatively correlated with the Λ8 values ($r^2$ = 0.73)
in the Rajang, and with a higher degradation signal observed in the upper Rajang, which drains
mineral soils with lower lignin levels (Fig. 4a). However, the Λ8 values are positively correlated
with the $(Ad/Al)v$ ratios ($r^2$ = 0.50) for the small rivers, except for the samples collected from the
Maludam in March 2017 (Fig. 4b). Such a distribution could be related to the grain size effect, as
illustrated in Fig. 4c and 4d. Of the sediments sampled here, the upper Rajang samples contain
the largest coarse fraction and the finest sediments were collected from the Maludam in March
2017. The $(Ad/Al)v$ ratios increase with decreasing mean size of the sediments in the small rivers.
Selective sorption of acid to aldehyde might affect the variation of the $(Ad/Al)v$ ratio in the small
river systems (Hernes et al., 2007). However, the relatively fresh condition of the OM in the
Maludam samples (in March 2017) might be related to the fluvial supply of fresh vegetation
during the flood season.





The syringyl and cinnamyl series are preferentially degraded when compared with the vallinyl
series, resulting in a decrease in the S/V and C/V ratios during lignin degradation (Goni et al.,
1995; Opsahl and Benner, 1995). Our samples show a negative linear relationship between the
S/V and (Ad/Al)v ratios in the Rajang samples ($r^2$ = 0.85; Fig. 5a). However, the variation of the
S/V and (Ad/Al)v ratios in the small rivers was limited, and a non-linear correlation is evident (Fig.
5b). Both correlations indicate that the decrease in the S/V ratios is linked to degradation, and
this suggests that we should be cautious when using S/V ratios for source evaluation in this study.
Previous studies demonstrated that lignin mineralization in humid tropical forest soils is
dominated by methoxyl-C mineralization under aerobic and fluctuating redox conditions (Hall
et al., 2015). Demethylation reduces the yield of methoxylated phenols (V and S phenols) but
does not affect P phenols. Therefore, the P/(S+V) ratio can be used as an indicator of lignin
transformation (Ditmar and Kattner, 2003). However, in this study the ratio of P/(S+V) in most
samples did not vary greatly (~0.2), and no clear trend was observed for the small rivers,
although there was a linear correlation between the P/(S+V) and (Ad/Al)v ratios ($r^2$ = 0.89).

**4.4 Impact of environmental parameters on lignin dynamics**
It is well explored that bulk organic matter composition and degradation are influenced by many
environmental factors such as climate, grain size, mineral composition, soil characteristics, land
use changes, logging, and biomass burning (Hernes et al., 2007; Gandois et al., 2014; Sun et al.,
2017; Thevenot et al., 2010). Most Southeast Asian peat-draining rivers are impacted by human
activities such as deforestation, urbanization and damming (Milliman and Farnsworth, 2011). The
PCA analysis revealed that the behavior of lignin in the Rajang is substantially different from that
in the three peat-draining rivers, and especially in the upper Rajang, which drains through a
mineral soil with low $\Lambda 8$ values and strong degradation (Figs 3 and 4).
In this study, the OC content increases with decreasing grain size, implying that fine particles,
with larger specific surface areas and rich in clay, contain more OM than coarser particles, as
reported previously (Sun et al., 2017). However, other studies found that coarse particulate OM





in the Amazon basin has a higher content of lignin phenols than fine POM, and also that coarse
POM is composed of fresher lignin derived from wood debris (Hedges et al., 1986). Nevertheless,
increasing $(Ad/Al)_V$ values were observed in the Rajang with increasing grain size, which suggests
that lignin associated with larger mineral particles is more strongly degraded. This observation
indicates the preferential preservation of lignin in finer-grained sediments, resulting from their
ability to provide better protection against further oxidative degradation (Killops and Killops,
2005). For the small river systems, the $(Ad/Al)_V$ ratios decrease with increasing grain size,
corresponding to the increasing $\Lambda 8$ values (Fig. 4a and b). Our observations of $(Ad/Al)_V$ values
are similar to the trends described by Keil et al. (1998) and Tesi et al. (2016), who found that lower
$(Ad/Al)_V$ values were present in the coarser fractions due to the less efficient processing of plant
remains prior to deposition. The sediments collected from the three small peat-draining rivers
(except samples from the Maludam in March, 2017) could have contained limited amounts of
plant debris, in which case fresh plant tissue would have been incorporated into the coarser
sediment fractions, leading to the low $(Ad/Al)_V$ values. However, the variation in $\Lambda 8$ values does
not support this speculation, and therefore we conclude that the selective sorption of acid to
aldehyde could explain the elevated $(Ad/Al)_V$ ratios recorded in the fine fraction. The different
grain-size effects on OCterr composition, as seen when comparing the Rajang with the small
rivers, suggests that there are other processes working on OCterr in these two systems, which
cause post-depositional changes in the OCterr characteristics.
Tropical soils are reported naturally poor in N and P, but some studies have shown that with
intensive management (land use/deforestation) they tend to become rich in recalcitrant
compounds, and recalcitrant substrates will continue to decompose, when the conditions
microbial preferred (Thevenot et al., 2010). In our study, we found a higher TN% in the small
rivers compared with the Rajang. A good correlation between $\Sigma 8$ and TN% was observed in all
systems, which might suggest a contribution from plant litter affecting both parameters (Fig. 6a).
However, the $(Ad/Al)_V$ ratios appear to be related to the C/N ratios, but with different slopes
obtained for the Rajang and the small rivers (Fig. 6b). This could be relative to the expected





impact of nitrogen on lignin degradation (Dignac et al., 2002; Thevenot et al., 2010). A high N
content will inhibit fungal lignin biodegradation (Fog, 1988; Osono and Takeda, 2001), and this
explains why slower degradation was observed in the small river systems in which higher TN%
were recorded. The exceptional data were collected during September 2017, which was a time
of saline water intrusion.
Large-scale land reclamation, including deforestation and urbanization, has taken place in
Southeast Asia over the past few decades (Miettinen et al., 2016). Logging activities have had a
significant influence on peat decomposition processes and the quality of organic matter inputs
(Hoscilo et al., 2011; Hooijer et al., 2012; Gandois et al., 2014). Gandois et al. (2013) reported an
increase in the N content at a deforested site and concluded that it was caused by an increase
in the microbial deposition of peat. The lignin yield ($\Sigma 8$) is closely correlated with the OC% in the
different rivers (Fig. 2). The highest yield was observed at the Maludam, which confirmed the
significant contribution of plant litter and better preservation due to the low pH and DO levels,
especially woody carbon. However, the relatively higher yield in the Rajang compared with the
other two disturbed peat-draining rivers (i.e., the Simunjan and Sebuyau) suggests an additional
source of lignin, which might implicate the addition of logging residue to the Rajang systems, as
proposed by Gandois et al. (2014).

**5 Conclusions**
We used sediment grain size data, TOC contents, the stable carbon isotopic composition of
organic matter, and lignin phenols concentrations to investigate the characteristics of OCterr in
a tropical peat-draining river system, as well as its fate and environmental controls. The depleted
$\delta^{13}C$ levels of all of the sediment samples demonstrates that contributions from C3 plants
dominated the OCterr in the study region. The lignin composition of the organic matter indicates
that the most important plant sources of organic matter were woody angiosperm C3 plants,
especially in the three small rivers. Our cluster and PCA analyses showed no distinct seasonal

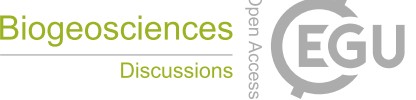

variations in the bulk and lignin compositional signatures in the study area, although the upper
Rajang receives contributions from mineral soils with unique lignin parameters and a coarser
grain size. Both the bulk organic matter parameters and the lignin compositions were indicated
to be correlated to the grain size of the riverbed sediments. The (Ad/Al)v ratios increased with
decreasing mean size of the sediments from the small rivers. Selective sorption of acid to
aldehyde might affect the variation of the (Ad/Al)v ratio in the small river systems. Our samples
show a negative linear relationship between the S/V and (Ad/Al)v ratios in the Rajang samples,
which implies that the decrease in S/V ratios is linked to degradation. The (Ad/Al)v ratios appear
to be related to the C/N ratio in the Rajang and the small rivers. A high N content will inhibit
fungal lignin biodegradation, which might explain the slower degradation observed in the small
river systems where a higher TN% was recorded. Most of the OCterr discharged from the Rajang
and small river systems was composed of woody angiosperm plants and the terrestrial organic
matter undergoes limited diagenetic alteration before deposition, and could potentially become
a significant regional carbon source to the atmosphere after extensive degradation.

*Author contributions.* YW, JZ, MM, and AM conceptualized the research project and planned the
field expeditions. JZ, MM, AM obtained research funding. JZ, KZ, JS, MM, MFM, EA and
AM collected samples and KZ and YW analyzed the samples. YW, KZ and JZ processed and
analyzed the data. All authors contributed to data interpretation and to the writing of the
manuscript.

*Competing interests.* The authors declare that they have no conflict of interest.

*Special issue statement.* This article is part of the special issue "Biogeochemical processes in
highly dynamic peat-draining rivers and estuaries in Borneo". It is not associated with a
conference.





**Acknowledgements**
The present research was kindly supported by the Newton-Ungku Omar Fund (NE/P020283/1),
the Natural Science Foundation of China (41530960), China Postdoctoral Science Foundation
(2018M630416), MOHE FRGS 15 Grant (FRGS/1/2015/WAB08/SWIN/02/1) and the SKLEC Open
Research Fund (SKLEC-KF201610). The authors would like to thank the Sarawak Forestry
Department and Sarawak Biodiversity Centre for permission to conduct collaborative research in
Sarawak waters under permit numbers NPW.907.4.4 (Jld.14)-161, Park Permit No WL83/2017, and
SBC-RA-0097-MM. Lukas Chin and the *SeaWonder* crew are acknowledged for their support
during the cruises. Dr. Zhuoyi Zhu, Ms. Lijun Qi, and the Marine Biogeochemistry Group are
especially acknowledged for their contribution and support during the sampling trips and
laboratory analysis.

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





**Table 1 Average values of bulk geochemical parameters for plants, soils, and sediments collected**
**from the study systems**

| Samples | Time | Mean Size (μm) | Clay% | Silt% | DO (mg/L) | pH | Salinity (‰) | OC (%) | TN (%) | Atomic C/N | δ¹³C (‰) |
|---|---|---|---|---|---|---|---|---|---|---|---|
| Angiosperm leaves & grasses (n=10) | 03/2017 | — | — | — | — | — | — | 48.53±2.86 | 1.65±0.64 | 40.44±18.95 | -31.1±2.5 |
| Angiosperm woods(n=5) | 03/2017 | — | — | — | — | — | — | 46.71±4.71 | 0.52±0.19 | 117.00±45.32 | -31.8±2.3 |
| Roots(n=3) | 03/2017 | — | — | — | — | — | — | 38.60±4.80 | 1.06±0.64 | 50.10±19.58 | -28.3±0.4 |
| Lower Rajang detritus(n=8) | 08/2016 | — | — | — | — | — | — | 40.76±13.69 | 0.94±0.35 | 47.21±13.03 | -29.9±2.1 |
| Sebuyau detritus(n=5) | 03/2017 | — | — | — | — | — | — | 30.63±15.00 | 0.73±0.20 | 53.39±31.68 | -28.1±2.0 |
| Simunjan detritus(n=4) | 03/2017 | — | — | — | — | — | — | 33.46±8.46 | 1.09±0.35 | 43.44±29.73 | -29.9±0.7 |
| Soil(n=8) | 09/2017 | — | — | — | — | — | — | 3.63±0.63 | 0.19±0.02 | 21.98±2.50 | -28.4±0.2 |
| Upper Rajang (n=4) | 08/2016 | 212.9±47.0 | 9.7±2.5 | 10.4±3.0 | 4.53±4.42 | 6.74±0.05 | 0 | 0.12±0.02 | 0.02±0.00 | 8.44±2.10 | -28.1±0.5 |
| Lower Rajang (n=16) | 08/2016 | 41.9±43.3 | 32.3±11.7 | 45.4±14.8 | 3.64±0.66 | 7.33±0.52 | 15.4±10.8 | 1.07±0.46 | 0.11±0.05 | 11.44±1.69 | -28.6±0.6 |
| Lower Rajang (n=5) | 03/2017 | 30.9±9.8 | 29.3±3.1 | 54.9±2.8 | 5.82±0.78 | 6.66±0.26 | 0.1±0.2 | 1.26±0.37 | 0.12±0.02 | 11.68±1.90 | -29.1±0.2 |
| Maludam (n=5) | 03/2017 | 9.3±2.3 | 39.6±2.7 | 59.3±2.0 | 3.24±2.24 | 4.93±1.71 | 7.2±10.0 | 2.22±0.69 | 0.20±0.05 | 12.83±1.80 | -27.4±0.6 |
| Maludam (n=2) | 09/2017 | 12.1 | 39.2 | 58.3 | 4.96 | 6.69 | 11.5 | 2.02 | 0.19 | 12.43 | -28.2 |
| Sebuyau (n=6) | 03/2017 | 24.6±18.5 | 31.6±6.5 | 58.8±8.3 | 3.07±1.92 | 5.40±5.48 | 5.5±6.5 | 2.37±0.69 | 0.16±0.03 | 17.37±4.56 | -27.8±0.3 |
| Sebuyau (n=5) | 09/2017 | 15.7±4.0 | 30.4±3.6 | 66.1±3.1 | 4.30±1.36 | 7.45±0.22 | 2.3±4.5 | 2.79±1.75 | 0.20±0.10 | 15.42±1.96 | -28.2±0.4 |
| Simunjan (n=6) | 03/2017 | 20.2±10.3 | 22.0±5.3 | 71.0±6.5 | 1.85±0.65 | 5.22±0.61 | 0 | 2.58±1.03 | 0.19±0.08 | 16.44±3.03 | -28.2±0.5 |
| Simunjan (n=6) | 09/2017 | 23.5±8.10 | 20.9±4.8 | 71.0±3.1 | 4.00±1.15 | 5.04±0.57 | 0 | 2.59±0.53 | 0.18±0.05 | 17.86±4.56 | -28.4±0.5 |




**Table 2** Average values of lignin phenols parameters for plants, soils, and sediments from the study systems (V: valinyl phenols; S: syringyl phenols; C: cinnamyl
phenols, P: p-hydroxyl phenols; DHBA: 3,5-dihydroxy benzoic acid; see the main text for definitions of $\Sigma 8$, $\Lambda 8$, Ad/Al, and LPVI)

| Samples | Time | $\Sigma 8$ (mg/10 g dw) | $\Lambda 8$ (mg/100 mg OC) | V | S | C | S/V | C/V | (Ad/Al)v | (Ad/Al)s | P/(V+S) | DHBA | DHBA/V | LPVI |
|---|---|---|---|---|---|---|---|---|---|---|---|---|---|---|
| **Angiosperm leaves & grasses (n=10)** | 03/2017 | 317.94 ±160.00 | 6.64±3.38 | 2.08±1.29 | 3.31±2.09 | 1.11±0.54 | 1.73±0.52 | 0.72±0.39 | 0.38±0.14 | 0.28±0.09 | 0.22±0.11 | 0.16±0.04 | 0.13±0.07 | 1420±910 |
| Angiosperm woods (n=5) | 03/2017 | 817.58±270.00 | 17.54±5.66 | 7.65±2.75 | 9.31±2.90 | 0.58±0.43 | 1.27±0.24 | 0.07±0.04 | 0.33±0.07 | 0.24±0.13 | 0.04±0.00 | 0.10±0.06 | 0.01±0.01 | 87±34 |
| Roots(n=3) | 03/2017 | 312.98±44.51 | 8.24±1.96 | 2.63±0.82 | 5.15±1.21 | 0.46±0.10 | 2.01±0.41 | 0.18±0.05 | 0.34±0.04 | 0.37±0.07 | 0.30±0.45 | 0.11±0.13 | 0.05±0.07 | 18±6 |
| Lower Rajang detritus(n=8) | 08/2016 | 418.98±151.87 | 11.57±6.47 | 5.40±2.60 | 5.35±3.73 | 0.86±0.56 | 0.89±0.24 | 0.18±0.10 | 0.35±0.09 | 0.27±0.12 | 0.24±0.10 | 0.26±0.18 | 0.08±0.13 | 10±55 |
| Sebuyau detritus(n=5) | 03/2017 | 638.41±373.55 | 20.39±3.15 | 9.63±2.01 | 9.70±2.29 | 1.05±0.64 | 1.04±0.33 | 0.11±0.05 | 0.34±0.13 | 0.37±0.09 | 0.15±0.09 | 0.16±0.11 | 0.02±0.01 | 85±34 |
| Simunjan detritus(n=4) | 03/2017 | 534.62±277.93 | 15.51±5.88 | 7.79±2.42 | 6.72±4.37 | 1.00±0.95 | 0.82±0.39 | 0.15±0.17 | 0.32±0.06 | 0.25±0.09 | 0.08±0.07 | 0.14±0.02 | 0.02±0.00 | 80±54 |
| Soil(n=8) | 09/2017 | 29.67±5.13 | 8.25±0.96 | 3.89±0.45 | 4.10±0.53 | 0.27±0.05 | 1.05±0.06 | 0.07±0.02 | 0.38±0.04 | 0.30±0.06 | 0.28±0.03 | 0.37±0.05 | 0.10±0.02 | 69±10 |
| Upper Rajang (n=4) | 08/2016 | 0.16±0.08 | 1.32±0.55 | 0.89±0.29 | 0.37±0.22 | 0.06±0.05 | 0.38±0.16 | 0.06±0.05 | 1.04±0.23 | 0.39±0.15 | 0.51±0.04 | 0.07±0.05 | 0.07±0.03 | 18±11 |
| Lower Rajang (n=16) | 08/2016 | 7.55±3.96 | 6.57±2.09 | 3.42±1.05 | 3.01±1.00 | 0.14±0.12 | 0.87±0.09 | 0.04±0.03 | 0.43±0.13 | 0.26±0.10 | 0.16±0.07 | 0.29±0.13 | 0.09±0.03 | 48±11 |
| Lower Rajang (n=5) | 03/2017 | 10.33±2.12 | 8.54±1.67 | 4.42±0.82 | 3.83±0.80 | 0.29±0.10 | 0.86±0.03 | 0.07±0.02 | 0.41±0.07 | 0.30±0.11 | 0.17±0.02 | 0.23±0.11 | 0.05±0.02 | 52±7 |
| Maludam (n=5) | 03/2017 | 14.21±6.66 | 6.21±1.40 | 3.62±0.99 | 2.53±0.46 | 0.07±0.05 | 0.71±0.07 | 0.02±0.02 | 0.58±0.18 | 0.30±0.18 | 0.20±0.01 | 0.44±0.13 | 0.12±0.04 | 33±6 |
| Maludam (n=2) | 09/2017 | 12.55 | 6.24 | 3.21 | 2.76 | 0.27 | 0.8 | 0.09 | 0.43 | 0.30 | 0.16 | 0.18 | 0.06 | 62 |





| | | | | | | | | | | | | | |
|---|---|---|---|---|---|---|---|---|---|---|---|---|---|
| Sebuyau (n=6) | 03/2017 | 18.02±7.07 | 7.75±2.10 | 4.50±1.33 | 3.12±0.82 | 0.13±0.108 | 0.70±0.05 | 0.03±0.02 | 0.47±0.07 | 0.34±0.06 | 0.17±0.04 | 0.33±0.08 | 0.08±0.01 | 33±6 |
| Sebuyau (n=5) | 09/2017 | 22.06±11.44 | 8.18±0.98 | 4.85±0.68 | 3.16±0.43 | 0.17±0.11 | 0.66±0.08 | 0.04±0.03 | 0.55±0.08 | 0.32±0.12 | 0.16±0.02 | 0.18±0.09 | 0.04±0.02 | 31±9 |
| Simunjan (n=6) | 03/2017 | 18.45±5.96 | 7.30±1.04 | 4.03±0.51 | 2.96±0.60 | 0.31±0.17 | 0.73±0.11 | 0.08±0.05 | 0.48±0.10 | 0.41±0.04 | 0.20±0.05 | 0.36±0.05 | 0.09±0.01 | 49±24 |
| Simunjan (n=6) | 09/2017 | 20.09±3.20 | 7.86±0.91 | 4.54±0.80 | 3.09±0.31 | 0.23±0.20 | 0.69±0.09 | 0.06±0.06 | 0.47±0.05 | 0.36±0.08 | 0.17±0.03 | 0.37±0.09 | 0.08±0.02 | 41±22 |










**Table 3 Comparison of bulk and lignin phenols parameters among river systems worldwide**

| Samples | Station | OC (%) | TN (%) | C/N | δ13C (‰) | Σ8 (mg/10g dw) | Σ8 (mg/100 mg OC) | S/V | C/V | (Ad/Al)v | (Ad/Al)s | P/(S+V) | DHBA/V | References |
|---|---|---|---|---|---|---|---|---|---|---|---|---|---|---|
| Amazon River | estuary | 0.13~1.44 | --- | --- | -29.4~-27.5 | 0.10~11.05 | 0.75~9.27 | 0.84~1.51 | 0.12~0.47 | 0.26~0.61 | 0.15~0.56 | --- | --- | 1 |
| Congo River | submerged delta | 0.8~4.2 2.1 | --- | 5.8~10.1 8.3 | -23.5~-19.0 | --- | 0.066~0.373 0.151±16% | 0.47~1.38 0.87±7% | 0.15~0.39 0.28±13% | 0.47~1.74 0.72±17% | 0.26~1.94 0.46±14% | --- | --- | 2 |
| Pichavaram River | estuary | --- | --- | 14.17±1.33 | -27.15±1.53 | --- | --- | 1.26±0.32 | 0.19±0.12 | 0.68±0.11 | 0.81±0.21 | 0.57±0.097 | --- | 3 |
| 35 Indian rivers | North group | 0.61±0.3 | 0.04±0.01 | 18.7±6.9 | -22.9±0.9 | 0.11±0.12 | 1.6±1.0 | 0.9±0.2 | 0.2±0.1 | 0.7±0.2 | --- | 0.4±0.2 | 0.3±0.2 | 4 |
| | South group | 2.3±0.6 | 0.12±0.03 | 19.8±4.1 | -26.3±0.8 | 1.7±0.5 | 6.7±2.8 | 1.5±0.5 | 0.3±0.1 | 0.5±0.1 | --- | 0.2±0.2 | 0.1±0.2 | |
| Kapuas River | whole basin | 0.55~14.20 | 0.05~0.55 | 11.0~34.8 | -30.39~-27.29 | --- | 0.13~3.70 | 0.34~1.18 | 0.28~1.40 | 0.71~2.01 | 0.72~2.12 | --- | --- | 5 |
| Rajang River | estuary | 1.12±0.50 | 0.12±0.03 | 11.57±1.72 | -28.6±0.60 | 7.55±3.96 | 6.57±2.09 | 0.87±0.09 | 0.04±0.03 | 0.43±0.13 | 0.26±0.10 | 0.16±0.07 | 0.09±0.03 | This research |
| Maludam River | estuary | 2.22±0.69 | 0.20±0.05 | 12.83±1.80 | -27.4±0.61 | 14.21±6.66 | 6.21±1.40 | 0.71±0.07 | 0.02±0.02 | 0.58±0.18 | 0.30±0.18 | 0.20±0.01 | 0.12±0.04 | |
| Sebuyau River | whole basin | 2.37±0.69 | 0.16±0.03 | 17.37±4.56 | -27.8±0.27 | 18.02±7.07 | 7.75±2.10 | 0.70±0.05 | 0.03±0.02 | 0.47±0.07 | 0.34±0.06 | 0.17±0.04 | 0.08±0.01 | |
| Simunjan River | whole basin | 2.58±1.03 | 0.19±0.08 | 16.44±3.03 | -28.2±0.48 | 18.45±5.96 | 7.30±1.04 | 0.73±0.1 | 0.08±0.05 | 0.48±0.10 | 0.41±0.04 | 0.20±0.05 | 0.09±0.01 | |





| | | | | | | | | | | | | | Ref |
|---|---|---|---|---|---|---|---|---|---|---|---|---|---|
| **Yangtze River** | whole basin | 0.64±0.06 | — | — | -25.0±0.1 | 3.60±0.18 | 5.66±0.33 | 1.16±0.05 | 0.37±0.01 | — | — | — | 6 |
| **Mississippi River** | estuary | 1.2±0.5 | 0.1±0.06 | 13.4±2.8 | -23.7±0.80 | — | 1.64±0.53 | 0.93±0.30 | 0.03±0.01 | 0.27±0.14 | 0.20±0.07 | — | 7 |
| **Lena River** | delta | 2.06±0.33 | — | 15.88±3.33 | — | 0.41±0.19 | 1.96±0.81 | 0.43±0.02 | 0.42±0.36 | 1.28±0.30 | 1.04±0.24 | 0.30±0.03 | 8 |
| **Pristine peat** | Brunei | 52.4 | 1.95 | 31.35 | -30.4±0.8 | — | 5.65 | 0.82 | 0.05 | 0.42±0.10 | 0.40±0.01 | 0.28±0.05 | 0.12 |
| **Disturbed peat** | Brunei | 50.95 | 2.09 | 28.44 | -29.5±0.6 | — | 10.29 | 0.97 | 0.05 | 0 | 0.40±0.01 | 0.22±0.1 | 0.07 | 9 |

References:   1. Sun S, Schefuß E, Mulitza S et al., 2017; 2. Holtvoeth J, Wagner T, Schubert C J. 2003; 3. Prasad M B K, Ramanathan A L. 2009; 4. Pradhan U K, Wu Y,
Shirodkar P V et al., 2014; 5.Loh P S, Chen C T A, Anshari G Z et al., 2012; 6. Li Z, Peterse F, Wu Y et al., 2015; 7. Bianchi T S, Mitra S, McKee B A. 2002; 8. Winterfeld M,
Goñi M, Just J et al., 2015; 9. Gandois L, Teisserenc R, Cobb A R et al., 2014.






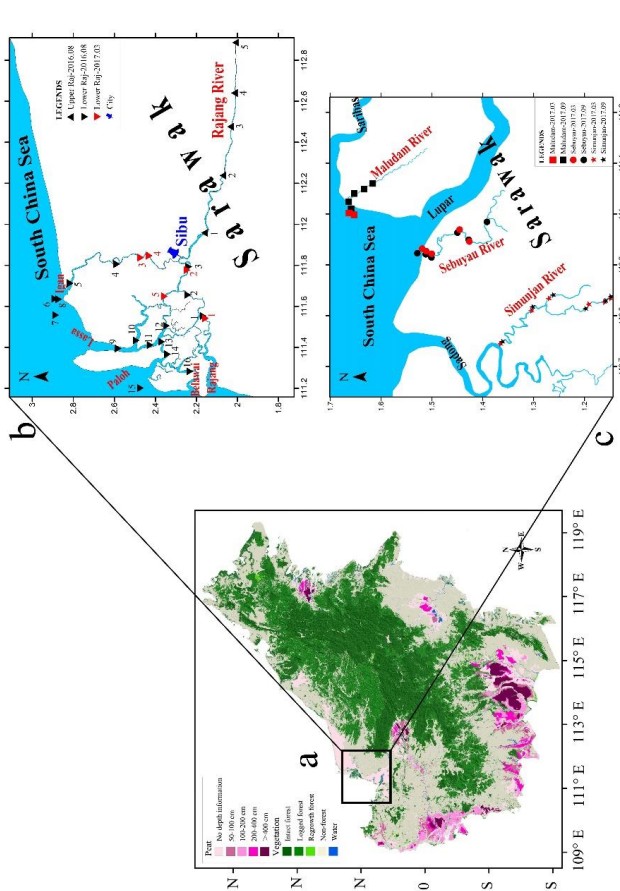

Figure 1 (a) Peat and vegetation distribution in the study region (modified from https://www.cifor.org/map/atlas/). (b) Sediment sampling sites along the Rajang River. The city of Sibu divides the river into upper and lower reaches. (c) Sediment sampling sites along the three small rivers. Locations of samples collected from the Maludam, Sebuyau, and Simunjan are indicated by squares, circles, and stars, respectively.





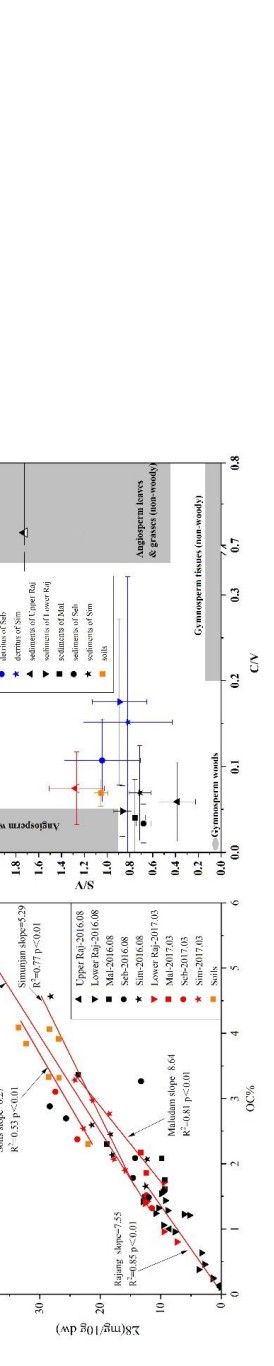



Figure 2 (a) Correlation of OC% with Σ8 among the various study systems. (b) Variations of S/V *versus* C/V of different samples from the study systems. Raj:


Rajang; Seb: Sebuyau; Sim: Simunjan; Mal: Maludam.





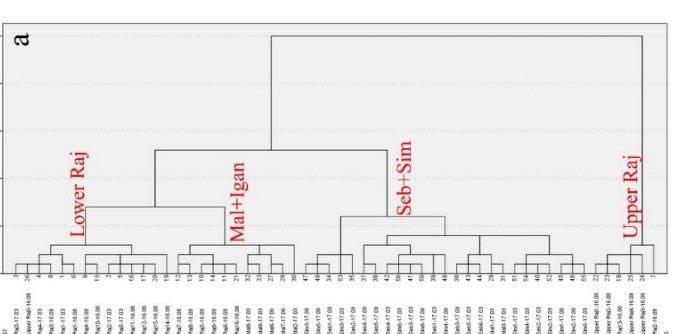

Figure 3 (a) Cluster analysis of the study systems based on bulk and lignin phenols parameters. (b) Plot of PCA results based on the distribution of scores 1
and 2. Raj: Rajang; Seb: Sebuyau; Sim: Simunjan; Mal: Maludam.





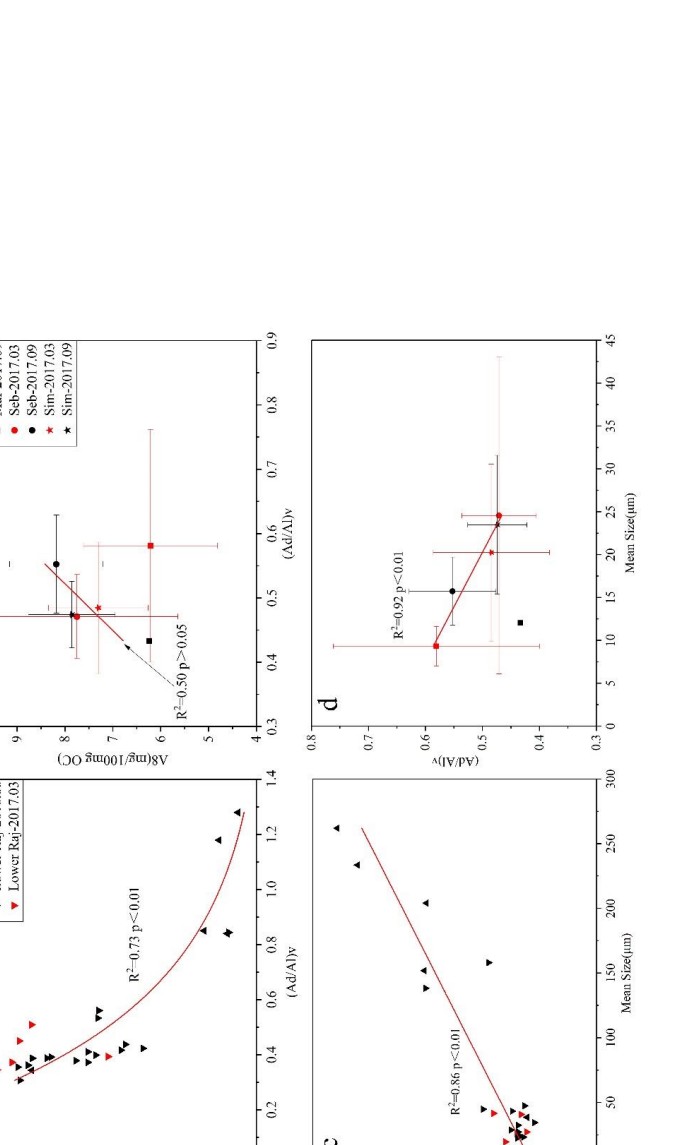



Figure 4 Variation in (Ad/Al)v with Λ8 values of sediments from (a) the Rajang and (b) the small river systems. Variation in (Ad/Al)v with mean sediment grain
size for (c) the Rajang and (d) the small river systems.





Figure 5 Relationship between (Ad/Al)v and S/V ratios based on average values of the various systems for (a) the Rajang and (b) the small river systems.






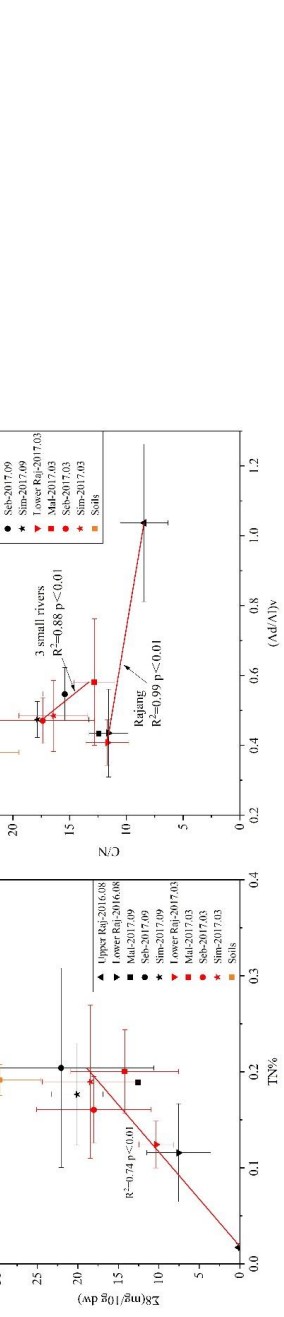

Figure 6 (a) Correlation of TN% with Σ8 based on average values of the study systems. (b) Correlation of (Ad/Al)$_v$ with C/N ratio based on average values of the study systems.