# Peer review of "Distribution and degradation of terrestrial organic matter in the # sediments of peat-draining rivers, Sarawak, Malaysian Borneo"

_Biogeosciences, 2019_

## Referee Comment (RC1) · Anonymous Referee #1 · 19 Apr 2019

Wu et al. presented a large dataset of organic carbon and lignin in the peat-draining river systems in Malaysian Borneo. Bulk properties and lignin phenols were used to elucidate the sources and degradation degree of organic matter in those systems. In my opinion, the dataset is valuable, and the manuscript is overall well-written. I only have some minor comments, hope this can help to further improve this manuscript.

1. Study region and sample collection: I suggest the authors add more background introduction and more sampling details. e.g., how the water discharge changes in those rivers monthly, and what were the discharges like during different sampling campaigns of each river? This may help to give the readers an overall impression of different

hydrological conditions; soils: were they all surface samples or depth profiles? How were they collected? How representable were they? 2. Chemical analysis: what is the deviation of lignin analysis? This is important, as it can help to understand whether the variation of lignin indices were its natural variation or analytical error. 3. Line 160: Please add reference for this equation. 4. Please add the information of what parameters were used in PCA and cluster analysis. 5. Lines 194-195: The highest values of . . . (30%-49%), this sentence reads odd, please revise. 6. Line 200: delete which before exhibited 7. Line 216: When describe correlations, the authors often use "good", however, it is not clear, what kind of relationship is "good", please define "good" (e.g., correlation coefficient R2 higher than 0.7), or use other words, e.g., significant or simply show R2 8. Line 238: do you mean which reflects the very low "P" value? 9. Line 254: I am a little confused here, do you mean that factor 2 showed close correlations with $\Sigma 8$ and OC%, and factor 1 showed close correlations with (Ad/Al)v and grain size? 10. Lines 265-267: Could you please explain more how higher values of $\Lambda 8$ is caused by human activities? 11. Line 272-275: I think this sentence could go to the conclusion 12. Line 278: The average $\delta 13C$ of vegetation samples seems lower than -28.5‰ please double-check this value. 13. Line 288-289: Do you mean close relationship between OC% and scores on factor 2, $\Sigma 8$ and scores on factors 2? 14. Line 296: I suggest the authors add the loading of the main parameters on the two components (factors). 15. Lines 315-318: Here the authors attributed the higher P/V ratio to the non-woody input for the additional lignin? Could you please explain more what is additional lignin? 16. Line 321: change "estimate" to "evaluate" 17. Lines 331-338: the (Ad/Al)v and $\Lambda 8$ relationship, in Rajang river, the authors used each single data point, but in the small rivers, the authors used average value? What if you do it the similar way, do you still see the correlation between (Ad/Al)v and $\Lambda 8$ or (Ad/Al)v and grain size in small rivers? The variation of (Ad/Al)v were much smaller in small rivers as compared to the Rajang river. In my opinion, grain size may not be the most important factor for (Ad/Al)v in small rivers as it also showed smaller variation. 18. Lines 339-345: again for the correlation between S/V and (Ad/Al)v in small rivers, what

if you use individual data point instead of average value? In Fig. 5b, it seems that the whole relation is because of Mal-2017.09, in other words, it seems that there was no significant relation between S/V and (Ad/Al)v in other two rivers. 19. Fig. 6b: for Rajang, there were only three points, this kind of relationship is artificial, in my opinion. Why not using single data point instead of average value? 20. Table 3: Please be careful with the effective number, e.g., C/N. 21. Table 3: The font of last line is different to others, please revise.

―――――――――――――――

---

## Short Comment (SC1) · 2 May 2019

I have read this paper with great interest as organic matter composition in tropical peat and peatland-draining rivers is not so well constrained compared to the Arctic counterparts. I have comments concerning two aspects of the paper for the authors to consider.

1. Influences of source vegetation on lignin composition should be considered. Line 121: Details of soil and plant sampling are not given. For instance, what is the depth of sampling for soil samples? What are the dominant plant species (trees, grasses, shrubs)? A distinct difference between tropical and (sub)arctic peatlands is that vegetation is dominated by woody species in the former versus by Sphagnum in the latter. This also explains the high abundance of lignin phenols in the studied river sediments (Line 265). Line 269: Again, I think the discussion of Ad/Al values should be put in the context of vegetation differences. Some grasses in the alpine grasslands of Qinghai-Tibetan Plateau are found to have high Ad/Al values in their roots, for instance (Zhu et al., 2019, Plant and Soil, doi: 10.1007/s11104-019-04035-8).

2. Statistics. Tables: Do errors represent standard errors or standard deviation? Is comparison of mean values tested by statistical analysis?

---

## Referee Comment (RC2) · David Burdige (Referee) · 26 Jul 2019

In this manuscript Wu et al present data on organic matter distribution in tropical peat-lands and rivers in Borneo, focusing on the use of lignin phenols to understand the transformations of terrestrial organic matter. The manuscript is reasonably well-written although there are a few places where the language and grammar could use some cleaning up. I have tried to indicate some of these places in my comments below, although the authors should probably have a native English-speaker go through the manuscript before sending the revised version back to the journal.

In the end I think this will be a useful contribution to the literature. However, I do have a

number of questions and comments about some aspects of their discussions and the interpretations of their data.

As a general comment, I have a concern that much of the literature data discussed here is (I believe) for POM suspended in rivers, and the data collected here is mainly from surface river sediments. As a result, there may be a little bit of a problem with these comparisons. Organic matter in sediments is processed differently than that which is suspended in the water column because of differences in the two settings in their solid:solution ratios as well as their exposure to oxygen. Therefore, depending on how "stable" the bottom sediments are (i.e., how often they are resuspended into the river, how long they spend in the river before being re-deposited, and then how long they remain deposited as sediments) these differences in environmental settings may impact the conclusions drawn in the comparisons presented here. I will try and point out specific places in the text where I think this needs to be looked at a bit more carefully.

Numbers in parentheses refer to line numbers.

1. (26) – add "phenols" after "aldehyde".

2. (28-9) – Here and in the conclusions (line 428) and the text (line 395) they refer to "slower degradation". I think they are really talking about material that shows less evidence of degradation since it's not clear to me how rates of processes can be inferred from any of the results presented here.

3. (38) - Add "is" before "derived".

4. (39) – I think "disturb" should be "disturbance".

5. (59) – While the presence of lignin is an indication of the presence of OCterr, its absence is not necessarily an indication of the lack of terrestrial organic matter, since highly degraded soil organic matter may be devoid of any apparent lignin. Perhaps this is a subtlety the authors don't want to get into, but I wonder if this is worth at least

mentioning?

6. (145) – A microwave "oven"?

7. (154) – I found this whole paragraph very confusing.

8. (160) – A reference or two, and some additional discussion is needed about this index. For example, what kinds of values do you typically see among different end-member materials, fresh versus degraded materials, etc.

9. (203) – Detritus samples are mentioned here for the first time. It's not clear to me from reading the methods section what they represent and how they were collected. This needs to be clarified.

10. (204-7) – By river "samples" do you mean river "sediments"? Also, I'm not sure I would be willing to say that there were differences between the $\delta$13C values in the peat-draining rivers versus the Rajang.

11. (209- ) – I think Sum8 is defined as per 10 g dw (not 10 mg). Otherwise, many of the values presented here suggest you have more lignin carbon than total material.

12. (257) - The discussion of results in Table 3 is a place where I have concerns about comparing literature data for suspended river POM and data here for bottom river sediments.

13. (270) – The transition here to discussing Arctic sediments is rather abrupt, and the point here is not clear to me.

14. (272) – This sentence ("This study . . .") says very little and seems out of place here.

15. (287) – The phrase ". . . sediment samples . . .phase" doesn't make sense to me.

16. (295-6) – I don't see how these relationships between $\delta$13C and Sum8 relate to Fig. 3.

17. (303-4) – This is a place where additional information about the LPVI index would be useful.

18. (329) – The correlation in Fig. 4b is very weak at best, and in fact, if you include the sample collected from the Maludam in March 2017 you would probably get just as strong an inverse correlation.

19. (331-8) – Why is there no discussion of the positive relationship between (Ad/Al)v and grain size in the Rajang (Fig. 4c)?

20. (343) – I'm not entirely convinced there is a non-linear relationship in Fig. 5b. Fitting a straight line through the data might show a correlation just as good as some of the linear correlations in Fig. 4.

21. (351) – What does "no clear trend" mean, especially in light of the linear correlation discussed on the next line.

22. (359-362) – I think that it would be good to provide a little more detail to support this statement. The fact that the PC analysis says that lignin degradation is different in the Rajang versus the peat-draining rivers is interesting, but it's not clear to me what that means, and what new information it is giving us about how terrestrial organic matter is processed in these systems.

23. (363- ) – This is another place where I have concerns about comparing literature data for suspended river POM and data here for bottom river sediments.

24. (387) – The phrase ". . . when the conditions microbial preferred . . ." doesn't make sense to me.

25. (392) – "relative" should be "related".

David Burdige

---

## Author Comment (AC1) · 21 Aug 2019

Revision notes to RC1 bg-2019-94 Distribution and degradation of terrestrial organic matter in the sediments of peat-draining rivers, Sarawak, Malaysian Borneo" by Ying Wu et al.

Comments: 1. Study region and sample collection: I suggest the authors add more background introduction and more sampling details. e.g., how the water discharge changes in those rivers monthly, and what were the discharges like during different sampling campaigns of each river? This may help to give the readers an overall impression of different hydrological conditions; soils: were they all surface samples or

depth profiles? How were they collected? How representative were they?

Reply: Thanks for the comments, we have revised as suggested, add the information about sediment and water discharge in seasonal patterns. However, there is almost no information available about those small rivers and also no information about monthly discharge from the Rajang. The detailed sampling information about soil samples are also added as suggested.

Revised: P5-P6 The Rajang River drainage basin covers an area of about 50,000 km2. Elevations exceed 2000 m and hill slopes are steep, generally in excess of 258 m in the interior highlands and 208 m in lower areas (Martin et al., 2018). The three small rivers (the Maludam, Simunjan, and Sebuyau) are blackwater rivers that draining extensive peatlands (Fig. 1). The drainage basin of the Maludam is about 91.4 km2 and the majority of the river is located in the Maludam National Park with 10m thick peat (Muller et al., 2015). The other two rivers are highly human disturbed with intensive oil palm and sago plantations. For the Rajang, it is separated into two parts by Sibu Town, upper reaches mainly drains mineral soils, while down reaches develops multiple distributary channels (e.g., the lower Rajang, Serendeng, Igan; Fig. 1). These channels are also surrounded by broad peatlands. It is reported that peat greater than 1m thick covered 50% of the delta plain (Staub et al., 2000). However, Deforestation and changing in land use are accelerating the peatland degradation (Fig. 1). More than 50% peatland (11% of the catchment size) in Rajang watershed has been occupied by industry plantation (e.g. oil palm) (Miettinen et al., 2016). Fishery, logging and timber processing are the traditional supports for local citizens (Miettinen et al., 2016). The climate of the study area is classified as tropical ever-wet, with average rainfall in excess of 3700 mm/year. The average monthly water discharge of the Rajiang is about 3600 m3/s, with peak discharge (∼25,000 m3/s) observed during the northeastern monsoon season (December to March; Staub et al., 2000). However, the amount of suspended sediments delivered from the Rajang basin to the delta plain demonstrated slightly variation (2.0MT/s dry season versus 2.2 MT/s wet season) but changed substantially about the amount of sediment delivered from the delta plain to the South China Sea (Staub et al., 2000). It is estimated that the annual sediment discharge of the Rajang was 30 Mt. The turbidity maximum in the lower Rajang channels occurred during the low or reduced discharge period. It is reported that up to 24 Mt of sediment was deposited in the delta front with preserved annual sediment layers at the order of one cm thick (Staub et al., 2000). The water discharge of the Maludam is quite low, only $4.4\pm0.6$ m3/s, from the 91.4 km2 catchment (Muller et al., 2015). The river length of Maludam is 33 km. For the Sebuyau and Simunjan, river length is 58 and 54 km, respectively (Martin et al., 2018). However, hydraulic information for these two rivers is largely unknown. The three sampling periods resembled the end of this northeastern monsoon (i.e., March, the end of the wettest season of the year) and were shortly before the beginning of the northeastern monsoon (i.e., August and September, the end of the drier season).

Comments: 2. Chemical analysis: what is the deviation of lignin analysis? This is important, as it can help to understand whether the variation of lignin indices were its natural variation or analytical error. Reply: Added as suggested. The deviation is less than 10%. Revised: P8 L183- Coefficients of analytical variation associated with phenols values were <10% based on replicate analysis of the same samples.

3. Line160: Please add reference for this equation. Reply: Added as suggested. Revised: P8L188- Since both ratios have been found to decrease with the preferential degradation of S and C relative to V phenols, lignin phenols vegetation index (LPVI) was developed to be an alternative approach to evaluate the original of various type of vegetations (Tareq et al., 2004; Thevenot et al., 2010):

4. Please add the information of what parameters were used in PCA and cluster analysis. Reply: Added as suggested Revised: P9L200- Multivariate statistical approaches such as principle component analysis (PCA) and cluster analysis (CA) are among the most widely used statistical methods in determining the significance of specific parameters (including OC%, TN%, mean grain size, clay% and silt%, total lignin phenols concentrations, DHBA and the ratios of vanillic acid to vanillin ((Ad/Al)V)) within a dataset (Pradhan et al., 2009).

5. Lines 194-195: The highest values of : (30%-49%), this sentence reads odd, please revise. Reply: Revised as suggested. Revised: P10L236: The highest values of OC were measured in plants samples and varied from 30%–49% (Table S2).

6. Line 200: delete which before exhibited Reply: Did as suggested. Revised: P10L241 Although nitrogen was enriched in the samples from the peat-draining rivers, they still had higher mean C/N values (15.8±3.7) compared with the lower Rajang (11.5±1.6) while vegetation samples, exhibited low N content and high C/N (C/N = 56±34).

7. Line 216: When describe correlations, the authors often use "good", however, it is not clear, what kind of relationship is "good", please define "good" (e.g., correlation coefficient R2 higher than 0.7), or use other words, e.g., significant or simply show R2 Reply: Yes, it is a good comment, revised as suggested through the whole manuscript. Revised: P11L259 There are correlations between ïĄŞ8 and OC% in each river (r2 > 0.5), with the slope decreasing in the order of Maludam > Simunjan > Sebuyau > Rajang (Fig. 2a). P14, L330- ; L350- The close correlation of factor 2 with OC% and ïĄŞ8 in the PCA suggests factor 2 relates to the source of the organic matter (Fig. 3), as also indicated by the strong correlation between OC% and ïĄŞ8 (r2: 0.53-0.85) (Fig. 2) For the Rajang, the LPVI values show a positive linear correlation with ïĄŇ8 concentrations (r2 = 0.56);

8. Line 238: do you mean which reflects the very low "P" value? Reply: Yes, it is. Revised: P12L280 The P/(V + S) ratio is low in the vegetation samples, except for the leaf samples (P/(V + S) = 0.22), which reflects the low P content in most vegetation (Table 2).

9. Line 254: I am a little confused here, do you mean that factor 2 showed close correlations with 8 and OC%, and factor 1 showed close correlations with (Ad/Al)v and grain size? Reply: Modified as suggested. Revised: P12 L294- Similar groupings are evident in the results of the PCA analysis, which was based on the distribution of factors 1 and 2 that represent total loadings of 45% and 32%, respectively (Fig. 3b). The PCA results implied that factor 1 showed close correlations with the (Ad/Al)v ratio and grain size while factor 2 showed a close correlation with ïĄŞ8 and OC%.

10. Lines 265-267: Could you please explain more how higher values of lig8 is caused by human activities?

Reply: The higher values of ïĄŇ8 found in our studied systems were linked to vegetation types (trees dominated) and partially caused by peat-draining and intense human activity near the watersheds (e.g. land use change and logging activities).

Revised: P13 L309- The higher values of ïĄŇ8 found in our studied systems were linked to vegetation types (trees dominated) (Zaccone et al., 2008) and partially caused by peat-draining and intense human activity near the watersheds (e.g. land use change and logging activities), as reported previously (Milliman and Farnsworth, 2011; Moore et al., 2013; Rieley et al., 2008). Much of the peatland neighboring the Simunjan and Sebuyau catchments has been changed to oil palm plantations (Martin et al., 2018). P17L417- The higher values of ïĄŞ8 and OC% were observed in Simunjan and Sebuyau, where land use and drainage observed. Usually land use and drainage of tropical peat will accelerate the loss of vegetation and OC degradation (Kononen, et al., 2016), here it may be explained by the high content of OC and lignin in oil palm, which is the major plantation in both regions.

11. Line 272-275: I think this sentence could go to the conclusion Reply: Modified as suggested. Revised: P20L488- Most of the OCterr discharged from the Rajang and small river systems was composed of woody angiosperm plants and the terrestrial organic matter undergoes limited diagenetic alteration before deposition, and could potentially become a significant regional carbon source to the atmosphere after extensive degradation. This study provides new insights into the amount of terrestrial OC preserved in the tropical delta region of southeastern Borneo, as well as into the biogeochemical transformation of OM from terrestrial source to marine sink across this region.

12. Line 278: The average 13C of vegetation samples seems lower than -28.5‰ please double-check this value. Reply: Yes, we modified and made it clear. Revised: P13L321- The depleted average ïΑď13C values (-31.8 ∼ −28.1‰ of our vegetation samples indicate an insignificant contribution from C4 plants in the study area (Gandois et al., 2014; Sun et al., 2017).

13. Line 288-289: Do you mean close relationship between OC% and scores on factor 2, 8 and scores on factors 2? Reply: We modified the sentence to make it clear as suggested. Revised: P14L330- The close correlation of factor 2 with OC% and ïΑŞ8 in the PCA suggests factor 2 relates to the source of the organic matter (Fig. 3), as also indicated by the strong correlation between OC% and ïΑŞ8 (r2: 0.53-0.85) (Fig. 2).

14. Line 296: I suggest the authors add the loading of the main parameters on the two components (factors). Reply: The loading of two components is 45% and 32%, modified as suggested. Revised: P14L334- Furthermore, the differences between the upper and lower Rajang are highlighted by the PCA results (score 1 represents 45% of the total loading while score 2 is 32%) and bulk parameters; i.e., the upper Rajang drains a mineral soil whereas peat is dominant in the delta region.

15. Lines 315-318: Here the authors attributed the higher P/V ratio to the non-woody input for the additional lignin? Could you please explain more what is additional lignin?

Reply: We check the information of plant samples, add some information for the potential contributions for higher P/V ratios observed in small river systems.

Revised: P15L362- Considering this, non-woody angiosperms are the most likely source of additional lignin. Combined the composition of P and V in plants samples listed in Table S2, we find some dominant species, e.g. Dipterocarpaceae, Bruguierag ymnorrhiza(L.) Poir., Elaeis guineensis Jacq. have a relatively higher P/V ration in their non-woody parts.

16. Line 321: change "estimate" to "evaluate" Reply: Done. Revised: P15L368-(Ad/Al)v ratios are often used to evaluate the degradation status of terrestrial OM.

17. Lines 331-338: the (Ad/Al)v and lig8 relationship, in Rajang river, the authors used each single data point, but in the small rivers, the authors used average value? What if you do it the similar way, do you still see the correlation between (Ad/Al)v and lig8 or (Ad/Al)v and grain size in small rivers? The variation of (Ad/Al)v were much smaller in small rivers as compared to the Rajang river. In my opinion, grain size may not be the most important factor for (Ad/Al)v in small rivers as it also showed smaller variation.

Reply: We totally agree these opinions, actually we tried both ways to interpret the data we have got during preparation. Due to the size of the Rajang basin, the scattered dots could be more suitable to reflect the variation in general, while the small rivers, all data shown limited variation, if we still prefer single scattered data, there will be no clear trend observed at all. We used the average values to track their relations with variable parameters, just in order to provide the clear systematic pattern among those data.

Revised: No corrections for this point.

18. Lines 339-345: again for the correlation between S/V and (Ad/Al)v in small rivers, what if you use individual data point instead of average value? In Fig. 5b, it seems that the whole relation is because of Mal-2017.09, in other words, it seems that there was no significant relation between S/V and (Ad/Al)v in other two rivers.

Reply: If we use single dots for the small rivers, there is weak relation between S/V andïijĹAd/AlïijĽv in each system, but the reverse trend of S/V withïijĹAd/AlïijĽv still existed, based on this point, we decided to evaluate the relationship with average values.

Revised: No corrections for this point.

19. Fig. 6b: for Rajang, there were only three points, this kind of relationship is artificial, in my opinion. Why not using single data point instead of average value?

Reply: ïïjŽIf we use all the individual data of RajangïïjŇwe could see there a reasonable relationship for the C/N and ïïjĹAd/AlïïjĽv ïïjĹr2=0.34ïïjĽïïjŇbut since the small river systems use average data to present, we choose the same principle for the Rajang system to keep it comparable within one figure.

Revised: No corrections for this point.

20. Table 3: Please be careful with the effective number, e.g., C/N. Reply: Revised properly. Revised: Please refer to the Table 3.

21. Table 3: The font of last line is different to others, please revise. Reply: Did as suggested. Revised: Please refer to the Table 3.

Please also note the supplement to this comment:
https://www.biogeosciences-discuss.net/bg-2019-94/bg-2019-94-AC1-supplement.pdf

[Figure]

**Supplement:**

[revised manuscript text omitted]

---

## Author Comment (AC2) · 21 Aug 2019

Revision notes to SC1 bg-2019-94 Distribution and degradation of terrestrial organic matter in the sediments of peat-draining rivers, Sarawak, Malaysian Borneo" by Ying Wu et al.

Comments: 1. Influences of source vegetation on lignin composition should be considered. Line121: Details of soil and plant sampling are not given. For instance, what is the depth of sampling for soil samples? What are the dominant plant species (trees, grasses, shrubs)? A distinct difference between tropical and (sub)arctic peatlands is that vegetation is dominated by woody species in the former versus by Sphagnum in

the latter. This also explains the high abundance of lignin phenols in the studied river sediments (Line 265). Line 269: Again, I think the discussion of Ad/Al values should be put in the context of vegetation differences. Some grasses in the alpine grasslands of Qinghai- Tibetan Plateau are found to have high Ad/Al values in their roots, for instance (Zhu et al., 2019, Plant and Soil, doi: 10.1007/s11104-019-04035-8).

Reply: Thanks for the great comment, following your suggestion, we explored the literature about the impact of sources of vegetation on lignin composition and the revision is quite benefited from it. The missing information of soil and plant sampling is added in 2.1 study region and sample collection section. The discussion of Ad/Al values in P13 is also revised by considering the vegetation differences as suggested.

Revised: P2 L26-: The selective sorption of acid relative to aldehyde phenols might explain the variations in the $(Ad/Al)v$ ratio. Elevated $(Ad/Al)V$ values observed from the Maludam's sediments may be also attributed to source plant variations. P6 L139- , L144-: The surface sediments were sampled at the middle stream of river using grab samplers from a small boat at each station and then 0 - 5 cm subsamples were collected and frozen ($-20°C$) until they were dried for subsequent analysis in the laboratory. Soil sampling was conducted at the same time along the Rajang river bank where the sites have minimal human disturbances and short soil cores were collected and mixed in situ as one composite sample for the depth of 0-10cm by getting rid of visible roots and detritus. The vegetation of tropical peat swamp forest is dominated by trees, e.g. the Anacardiaceae, Annonaceae and Euphobiaceae etc. (Page et al., 2006). Fresh, typical vegetations (listed in Table S2) were separately collected by leave, stem and roots, some detritus, which floating at the surface layer of the rivers were also collected for the comparison study. P13L315-: The $(Ad/Al)V$ values of the sediments sampled here are comparable to fresh and only low to medium oxidized. Elevated $(Ad/Al)V$ values observed from the Maludam's sediments (March, 2017) may be also attributed to source plant variations as observed in other study case (Zhu et al., 2019). Comments: 2. Statistics. Tables: Do errors represent standard errors or

standard deviation? Is comparison of mean values tested by statistical analysis?

Reply: Errors listed in the tables represent standard deviations, the comparison of mean values tested in Figures has been tested by statistical analysis, p values are listed for the information. We add some words in statistics part for the clearness.

Revised: P9 L207-: Errors listed in tables represent standard deviations for the analytical data. Differences and correlations were evaluated as significant at the level of $p < 0.01$.

References: Page, S.E., Reiley, J.O., and Wust, R.: Lowland tropical peatland of Southeast Asia. In: Peatlands: Evolution and Records of Environmental and Climate Changes (eds., by Martini, I.P., etc.) Elsevier, pp145-171, 2006. Zaccone, C., Said-Pullicino, D., Gigliotti, G., Miano, T.M.: Diagenetic trends in the phenolic constituents of Sphagnum-dominated peat and its corresponding humic acid fraction, Org. Geochem., 39, 830-838, 2008. Zhu, S., Dai, G., MA, T., Chen, L., Chen, D., Lu, X….Feng X.J.: Distribution of lignin phenols in comparison with plant-derived lipids in the alpine versus temperate grasslands soils, Plant and Soil, 1-14, 2019.

Please also note the supplement to this comment:
https://www.biogeosciences-discuss.net/bg-2019-94/bg-2019-94-AC2-supplement.pdf

---

## Author Comment (AC3) · 21 Aug 2019

Revision notes to RC2 bg-2019-94 Distribution and degradation of terrestrial organic matter in the sediments of peat-draining rivers, Sarawak, Malaysian Borneo" by Ying Wu et al.

Comments: As a general comment, I have a concern that much of the literature data discussed here is (I believe) for POM suspended in rivers, and the data collected here is mainly from surface river sediments. As a result, there may be a little bit of a problem with these comparisons. Organic matter in sediments is processed differently than that which is suspended in the water column because of differences in the two settings in

their solid:solution ratios as well as their exposure to oxygen. Therefore, depending on how "stable" the bottom sediments are (i.e., how often they are resuspended into the river, how long they spend in the river before being re-deposited, and then how long they remain deposited as sediments) these differences in environmental settings may impact the conclusions drawn in the comparisons presented here. I will try and point out specific places in the text where I think this needs to be looked at a bit more carefully.

Reply: Many thanks for the good suggestion. Based on the limited available information, the Rajang Delta is still aggregation and the annual sediment rate of delta front is about 1 cm (Staub et al., 2000), which makes the seasonal comparison reasonable, but due to abnormal climate during two seasons, we could not make it. Resuspension and re-deposited process might be observed in the lower Rajang during dry season, where TMZ dominated, but it is related to the tide process, only one or two stations are covered. We add these information in the sampling section for the background information. For the comparison study in the manuscript e.g. Table 3, only the sediment samples were considered for the further comparison study. In Discussion 4.4 section, we deleted the comparison with suspended samples as suggested.

Revised: P6 L125- However, the amount of suspended sediments delivered from the Rajang basin to the delta plain demonstrated slightly variation (2.0MT/s dry season versus 2.2 MT/s wet season) but changed substantially about the amount of sediment delivered from the delta plain to the South China Sea (Staub et al., 2000). It is estimated that the annual sediment discharge of the Rajang is 30 Mt. The turbidity maximum in the lower Rajang channels occurred during the low or reduced discharge period. It is reported that up to 24 Mt of sediment is deposited in the delta front with preserved annual sediment layers at the order of one cm thick (Staub et al., 2000). P17L422- In this study, the OC content increases with decreasing grain size, implying that fine sediments, with larger specific surface areas and rich in clay, contain more OM than coarser sediments, as reported previously (Sun et al., 2017). Increasing (Ad/Al)V

values were observed in the Rajang with increasing grain size, which suggests that lignin associated with larger mineral particles is more strongly degraded.

Comments: 1. (26) – add "phenols" after "aldehyde". Reply: Modified as suggested. Revised: P2L26- The selective sorption of acid relative to aldehyde phenols might explain the variations in the (Ad/Al)v ratio.

Comments: 2. (28-9) – Here and in the conclusions (line 428) and the text (line 395) they refer to "slower degradation". I think they are really talking about material that shows less evidence of degradation since it's not clear to me how rates of processes can be inferred from any of the results presented here. Reply: Sorry for the confusing, we modified it in the revised manuscript to make it clear. Revised: P2L30- In small rivers, a quick decline of C/N ratios responses to the slower modification of (Ad/Al)v ratio by the meant of better preservation of lignin phenols. P18L450- Quicker decline of C/N ratios related to slower lignin degradation in small rivers, this could be related to the expected impact of nitrogen on lignin degradation (Dignac et al., 2002; Thevenot et al., 2010). P20L486- A high N content will inhibit fungal lignin biodegradation, which might explain higher lignin phenols with moderate degraded process observed in the small river systems where a higher TN% was recorded.

3. (38) - Add "is" before "derived". Reply: Modified as suggested. Revised: P2L40- It is reported that about 77% of the carbon stored in all tropical peatlands is derived from Southeast Asia.

4. (39) – I think "disturb" should be "disturbance". Reply: Modified as suggested. Revised: P3L42- However, increasing anthropogenic disturbance in the form of land use change, drainage and biomass burning are converting this peat into a globally significant source of atmospheric carbon dioxide.

5. (59) – While the presence of lignin is an indication of the presence of OCterr, its absence is not necessarily an indication of the lack of terrestrial organic matter, since highly degraded soil organic matter may be devoid of any apparent lignin. Perhaps

this is a subtlety the authors don't want to get into, but I wonder if this is worth at least mentioning? Reply: Yes, it is true, we modified the original sentence to avoid misleading. Revised: P3L62- Lignin, which constitutes up to 30% of vascular plant biomass, is a unique biomarker of OCterr although highly degraded soil organic matter, as another important contributor to OCterr, may be devoid of any apparent lignin.

6. (145) – A microwave "oven"? Reply: It should be microwave digestion system, Modified. Revised: P8L172- Briefly, the powdered samples were weighed and placed in O2 free Teflon-lined vessels, and digested in a microwave digestion system (CEM MARS5) at 150°C for 90 min (Goñi and Montgomery, 2000).

7. (154) – I found this whole paragraph very confusing.

Reply: We followed the suggestion, modify the paragraph, summarize variable parameters of lignin phenols, which will be applied in the following discussions.

Revised: P8L186- Ratios of syringyl-to-vanillyl phenols (S/V) and cinnamyl-to-vanillyl phenols (C/V) are often used to indicate the relative contribution of angiosperm and non-woody tissues versus gymnosperm wood, respectively (Hedges and Mann, 1979). Since both ratios have been found to decrease with the preferential degradation of S and C relative to V phenols, lignin phenols vegetation index (LPVI) was developed to be an alternative approach to evaluate the original of various type of vegetations (Tareq et al., 2004; Thevenot et al., 2010): Lignin phenols vegetation index (LPVI) = [{S(S + 1)/(V + 1) + 1} × {C(C + 1)/(V + 1) + 1}] The ratio of P/(V+S) may reflect the diagenetic state of lignin when the other sources of P phenols (such as protein and tannin) are relatively constant (Dittmar and Lara 2001). The acid-to-aldehyde (Ad/Al) ratios of V and S phenols are often used to indicate lignin degradation and increases with increasing lignin oxidation (Otto and Simpson 2006).

8. (160) – A reference or two, and some additional discussion is needed about this index. For example, what kinds of values do you typically see among different end-member materials, fresh versus degraded materials, etc.

Reply: Add the reference as suggested. Revised: P8L188- Since both ratios have been found to decrease with the preferential degradation of S and C relative to V phenols, lignin phenols vegetation index (LPVI) was developed to be an alternative approach to evaluate the original of various type of vegetations (Tareq et al., 2004; Thevenot et al., 2010) P14L346- This finding is further confirmed by the LPVI values (Gymnosperm woods: 1, non-woody Gymnosperm tissues, 3-27; Angiosperm woods: 67-415; non Angiosperm tissues: 176-2782),

9. (203) – Detritus samples are mentioned here for the first time. It's not clear to me from reading the methods section what they represent and how they were collected. This needs to be clarified. Reply: We add the information of detritus samples collection in sampling section as suggested. Revised: P7L146- Fresh, typical vegetations (listed in Table S2) were separately collected by leave, stem and roots, some detritus, which floating at the surface layer of the rivers were also collected for the comparison study.

10. (204-7) – By river "samples" do you mean river "sediments"? Also, I'm not sure I would be willing to say that there were differences between the 13C values in the peat-draining rivers versus the Rajang. Reply: Yes, river samples mean riverine sediments. After double checked our data of stable isotopes of sediments, we made the correction to make it better precise. Revised: P10L247- The isotope ratios of the peat-draining river's sediments (average ïА̧ď13C varied at -28.2 â̌Ť -27.4‰ were comparable with the Rajang's (average ïА̧ď13C = −28.6±0.6‰ (Tab. 3).

11. (209- ) – I think Sum8 is defined as per 10 g dw (not 10 mg). Otherwise, many of the values presented here suggest you have more lignin carbon than total material. Reply: Sorry for the mistake, we modified as suggested. Revised: P11L253- except for the lignin yield (ïА̧Ş8), which is the sum of C + S + V and is expressed as mg 10 g dw−1, and are presented in Fig. 2 as well as Tables 2 and S1-3. The highest yields were measured in the vegetation samples (300–900 mg 10 g dw−1). The lignin yield from the soil samples and the three small rivers (average of ∼30 mg 10 g dw−1) is also higher than that from the Rajang samples (average of <10 mg 10 g dw−1), with the

lowest value observed in the upper Rajang (0.16 mg 10 g dw–1; Table 2).

12. (257) - The discussion of results in Table 3 is a place where I have concerns about comparing literature data for suspended river POM and data here for bottom river sediments. Reply: All the information we summarized in Table 3 is about sediments samples no POM samples at all, we revised the sentence to make it clear. Revised: P13L302- Table 3 summarizes the distribution of bulk and lignin parameters of sediments from typical systems worldwide.

13. (270) – The transition here to discussing Arctic sediments is rather abrupt, and the point here is not clear to me. Reply: Thanks for the comments, we moved the Arctic sediments information as suggested. Revised: P13L314- The terrigenous OM has been affected by diagenesis, as (Ad/Al)V varies markedly among the different systems (Table 3). The (Ad/Al)V values of the sediments sampled here are comparable to fresh and only low to medium oxidized. Elevated (Ad/Al)V values observed from the Maludam's sediments (March, 2017) may be also attributed to source plant variations as observed in other study case (Zhu et al., 2019).

14. (272) – This sentence ("This study : : :") says very little and seems out of place here. Reply: Deleted as suggested.

15. (287) – The phrase ": : : sediment samples : : :phase" doesn't make sense to me. Reply: Thanks for the comment, we modified the sentences to be better presented. Revised: P14L327- Previous study reported that the sediment load from the basin to the delta was no seasonal pattern, combined with comparable precipitations during our two sampling seasons, our observations matched it (Martin et al., 2018; Staub et al., 2000).

16. (295-6) – I don't see how these relationships between 13C and Sum8 relate to Fig. 3. Reply: Sorry for the misleading, the plot of 13C and Sum8 was presented at Fig. 2. Revised: P14L330- The close correlation of factor 2 with OC% and ïAŞ8 in the PCA suggests factor 2 relates to the source of the organic matter (Fig. 3), as also indicated

by the strong correlation between OC% and ïĄŞ8 (r2: 0.53-0.85) (Fig. 2).

17. (303-4) – This is a place where additional information about the LPVI index would be useful. Reply: Modified as suggested. Revised: P14L346- This finding is further confirmed by the LPVI values (Gymnosperm woods: 1, non-woody Gymnosperm tissues, 3-27; Angiosperm woods: 67-415; non Angiosperm tissues: 176-2782),

18. (329) – The correlation in Fig. 4b is very weak at best, and in fact, if you include the sample collected from the Maludam in March 2017 you would probably get just as strong an inverse correlation. Reply: Yes, thanks for your comments, we redrew the figure and edited the sentences to make it more precise. Revised: P15L376- However, the ïĄŇ8 values with (Ad/Al)v ratios was not so significant in the small river systems as we expected, partially resulting from the variation of (Ad/Al)v also could be vegetation sources controlled (Fig. 4b).

19. (331-8) – Why is there no discussion of the positive relationship between (Ad/Al)v and grain size in the Rajang (Fig. 4c)? Reply: Added as suggested. Revised: P16L380- Of the sediments sampled here, the upper Rajang samples contain the largest coarse fraction and the finest sediments were collected from the Maludam in March 2017. The (Ad/Al)v ratios increase with increasing coarse fraction of the sediments in the Rajang, which is typically observed in other systems (Bianchi et al., 2002; Li eta l., 2015; Sun et al., 2017) (Fig. 4c).

20. (343) – I'm not entirely convinced there is a non-linear relationship in Fig. 5b. Fitting a straight line through the data might show a correlation just as good as some of the linear correlations in Fig. 4. Reply: We tested both linear and non-linear relationship for Fig 5b, the linear correlation has poor values for the fitting curve, r2=0.23ïijŇp=0.19ïijŇthat is the reason we chose nonlinear correlation to fit curve. Revised: No edited here.

21. (351) – What does "no clear trend" mean, especially in light of the linear correlation discussed on the next line.

Reply: Sorry for the confusion, it means good correlation among all samples, but no clear trend with small rivers samples, we corrected it as suggested.

Revised: P16 L401- However, in this study the ratio of P/(S+V) in most sediment samples did not vary greatly (∼0.2). Although there was a linear correlation between the P/(S+V) and (Ad/Al)v ratios among all the sediments (r2 = 0.89), no clear trend was observed for the small rivers, which may suggest both parameter's more links to source instead of diagenetic process in these systems.

22. (359-362) – I think that it would be good to provide a little more detail to support this statement. The fact that the PC analysis says that lignin degradation is different in the Rajang versus the peat-draining rivers is interesting, but it's not clear to me what that means, and what new information it is giving us about how terrestrial organic matter is processed in these systems.

Reply: Thanks for the comments, as a starting paragraph of whole section, we add some general introduction of the differences of TOM in the Rajang and peat-draining rivers, more detailed information are presented in the following paragraphs.

Revised: P17L415- since it was recently shown that lignin could decompose as fast as litter bulk carbon in mineral soils (Duboc et al., 2014). In the delta region, most parameters were quite comparable, except ïAŞ8 and OC% (Table S1). The higher values of ïAŞ8 and OC% were observed in Simunjan and Sebuyau, where land use and drainage observed. Usually land use and drainage of tropical peat will accelerate the loss of vegetation and OC degradation (Kononen, et al., 2016), here it may be explained by the high content of OC and lignin in oil palm, which is the major plantation in both regions.

23. (363- ) – This is another place where I have concerns about comparing literature data for suspended river POM and data here for bottom river sediments. Reply: The comparison with Sun et al., 2017, it refers to sediment samples only, but the comparison with Hedges (1986) was SPM samples, we revised it accordingly to make it more

precise. Revised: P17L423- In this study, the OC content increases with decreasing grain size, implying that fine sediments, with larger specific surface areas and rich in clay, contain more OM than coarser sediments, as reported previously (Sun et al., 2017). Increasing (Ad/Al)V values were observed in the Rajang with increasing grain size, which suggests that lignin associated with larger mineral particles is more strongly degraded.

24. (387) – The phrase ": : : when the conditions microbial preferred : : :" doesn't make sense to me. Reply: Modified as suggested to make it clear. Revised: P18L443- since nitrogen content tends to stimulates decomposition of low-lignin litter by decomposer microbes, but usually decrease the activity of lignolytic enzymes and inhibit decomposition of high-lignin litter (Knorr, et al., 2005; Thevenot et al., 2010).

25. (392) – "relative" should be "related". Reply: Modified as suggested. Revised: P18L452- Quicker decline of C/N ratios related to slower lignin degradation in small rivers, this could be related to the expected impact of nitrogen on lignin degradation (Dignac et al., 2002; Thevenot et al., 2010).

Please also note the supplement to this comment:
https://www.biogeosciences-discuss.net/bg-2019-94/bg-2019-94-AC3-supplement.pdf
* * *

---

## Author Response (AR2)

Revision notes to RRC1

bg-2019-94

Distribution and degradation of terrestrial organic matter in the sediments of peat-draining rivers, Sarawak, Malaysian Borneo" by Ying Wu et al.

1. *Comments: Line 357: What is additional lignin? I am still not clear.*

Reply:

Thanks for the comments, we modified the paragraph as suggested, make it more clear and fluent. See L354-360.

Revised:

All P/V values from the samples (0.13–0.28) are higher than the average P/V ratio of wood (0.05) but similar to the range observed for leaves (0.16–6.9; Hedges et al., 1986). Considering this, some non-woody angiosperms are the most likely source of high P phenols in the small rivers. Combined the composition of P and V in plants samples listed in Table S2, we find some dominant species, e.g. *Dipterocarpaceae, Bruguierag ymnorrhiza*(L.) *Poir., Elaeis guineensis Jacq.* have a relatively higher P/V ratios in their non-woody parts.

2. *Comments: Fig. 4b and Fig. 4d: I can understand that the authors try to find some trend between (Ad/Al)v and Λ8, and (Ad/Al)v and mean grain size, however, in my opinion, they don't have to be correlated. If there is no correlation, then there is. The samples were so scattered (as shown by the large error bar), I don't think this kind of data presentation is convincing.*

Reply: Accepted, we modified the Fig 4d as suggested, and related descriptions in various paragraphs have been revised accordingly. See L373-375, L379-380, L422-424.

Revised:

In additional, such a distribution could be related to the grain size effect, as illustrated in the Rajang with high correlation (Fig. 4c) and not so convincing but with a certain trend in small rivers (Fig. 4d).

The variation of (Ad/Al)v ratios with mean size of the sediments in the small rivers is not so convincing as the Rajang (Fig. 4d).

For the small river systems, the $(Ad/Al)_v$ ratios inattentively decrease with increasing grain size, corresponding to the increasing $\Sigma 8$ values (Fig. 4b and 4d). Such kind of trends have been described by Keil et al. (1998) and Tesi et al. (2016)

[Figure]

3. *Comments: Fig. 5b: I am glad to see the response, but why the authors not presenting them in this way? If there is still reverse correlation in each river, why not showing them? In my opinion, if there is no correlation between S/V and (Ad/Al)v for all the small river samples, but there are some correlations in each river, then this might indicate that there is inter-basin variability, and this kind of correlation was basin dependent. The current Fig. 5b is not convincing, I suggest the authors present it in the same way as they do in the response.*

Reply: Accepted, we revised the Fig 5b with scattering dots and modified the sentence in the text. See L387-388.

Revised:
However, the variation of the S/V and (Ad/Al)v ratios in the small rivers is limited, with a scattering decrease trend (Fig. 5b).

[Figure]

4. *Comments: Fig. 6b: Again, if there is correlation, why not simply showing it? One way you can do is to mentioned that that C/N and (Ad/Al)v were significantly correlated with R2 = 0.34, then telling the readers for what reason, average values were presented.*

Reply: Revised as suggested, please refer to L 443-446.

Revised:
The relation of (Ad/Al)v ratios with C/N ratios of the Rajang appears correlated ($r^2$= 0.34). For the comparison, average values were applied to two systems, we found the average (Ad/Al)v ratios had certain correlation with the average C/N ratios, but with different slopes for the Rajang and the small rivers (Fig. 6b).